# Drastic increase in the magnitude of very rare summer-mean vapor pressure deficit extremes

Mauro Hermann [1,2] ✉, Heini Wernli [1] & Matthias Röthlisberger [1,3]

Summers with extremely high vapor pressure deficit contribute to crop losses, ecosystem damages, and wildfires. Here, we identify very rare summer vapor pressure deficit extremes globally in reanalysis data and climate model simulations, and quantify the contributions of temperature and atmospheric moisture anomalies to their intensity. The simulations agree with reanalysis data regarding these physical characteristics of historic vapor pressure deficit extremes, and show a +33/+28% increase in their intensity in the northern/ southern mid-latitudes over this century. About half of this drastic increase in the magnitude of extreme vapor pressure deficit anomalies is due to climate warming, since this quantity depends exponentially on temperature. Further contributing factors are increasing temperature variability (e.g., in Europe) and the expansion of soil moisture-limited regions. This study shows that to avoid amplified impacts of future vapor pressure deficit extremes, ecosystems and crops must become more resilient not only to an increasing mean vapor pressure deficit, but additionally also to larger seasonal anomalies of this quantity.

Recent examples of high-impact summertime weather and climate extremes such as extensive forest fires[1,2], record-shattering heat waves[3–7], and intense drought[6–8], have become a hallmark of early twenty-first century climate change, causing US$ billions of insured losses every year[9]. A connecting element of these high-impact events are extremely large values of vapor pressure deficit (VPD), also referred to as air dryness or atmospheric water demand, which measures the difference between saturation ($e_s$) and actual water vapor pressure ($e_a$) in air[10], i.e., VPD = $e_s - e_a$.

Extreme heat often relates to large VPD[11], due to the exponential dependence of $e_s$ on temperature[12], and meteorological drought typically reduces the atmospheric moisture availability[13–15]. Thus VPD combines the meteorological variables temperature ($T$) and atmospheric moisture ($q$) in a way that is particularly relevant for ecosystems, e.g., forest carbon uptake and mortality[6,16–20], wildfires[21–23], and agricultural crop yields[24–26]. In the context of plant functioning and vegetation dynamics, longer-term extremes, e.g., of seasonal mean

VPD in summer, are of central importance due to the sessile and partly long-lived nature of plant organisms[27]. Moreover, on the seasonal timescale, positive land-vegetation-atmosphere feedbacks can further amplify large VPD as a result of soil desiccation and reduced plant evapotranspiration[28–30]. A summer VPD extreme, i.e., an unusually large seasonal VPD anomaly in summer, can therefore arise from exceptionally large $T$ and/or low $q$, which are linked to land-atmosphere coupling[28,31] and circulation anomalies[14].

Previous studies attributed tremendous socioeconomic consequences to recent summer VPD extremes (see affected regions in Fig. 1a): Two extensive VPD extremes in North America in 1988 and 2012 were associated with nation-wide agricultural disasters of $30–40 billion in economic losses[32,33]. The summer VPD extreme in 2011 was the most intense in the northern mid-latitudes (mean VPD anomaly of 0.78 kPa as per our definition, see "Methods"; Fig. 1b) and was directly linked to a record-extensive wildfire season in the southern U. S.[21]. Furthermore, cattle herds were diminished or forced to migrate north,

[1]Institute for Atmospheric and Climate Science (IAC), ETH Zürich, CH-8092 Zurich, Switzerland. [2]Present address: SRF Meteo, Swiss Radio and Television (SRF), CH-8052 Zurich, Switzerland. [3]Present address: Swiss Mobiliar Insurance, CH-3001 Bern, Switzerland. ✉e-mail: hermann.ethz@gmail.com

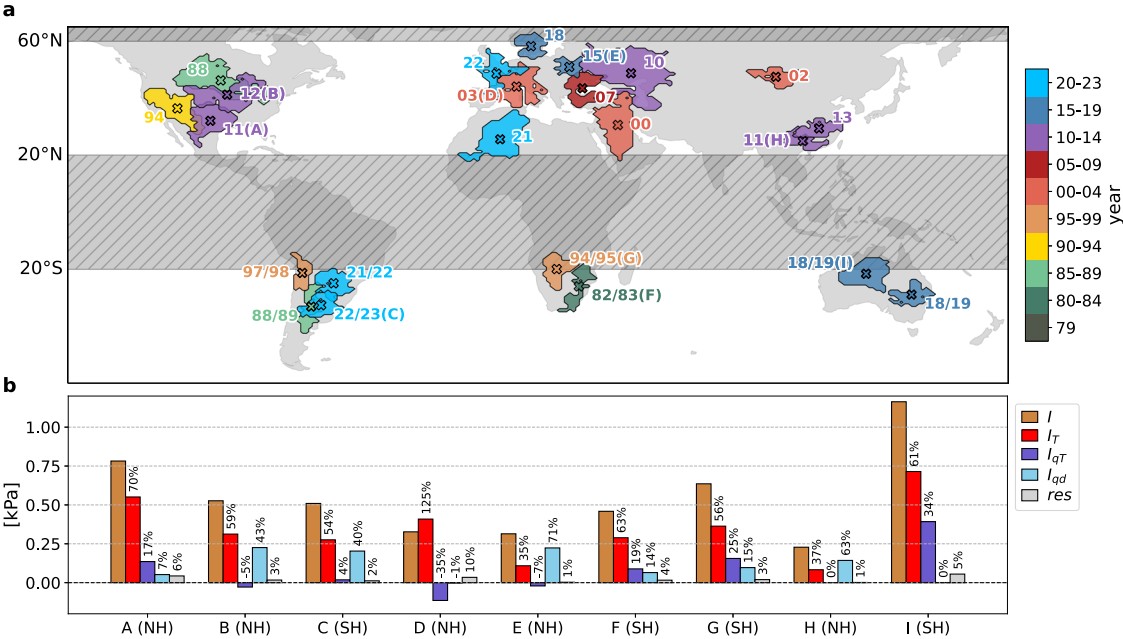

**Fig. 1 | Selected summer vapor pressure deficit (VPD) extremes in 1979–2023 and the meteorological contributions to their intensity. a** Selected summer VPD extreme objects, see "Methods" for details of the event identification. Each object is colored according to its year of occurrence and labeled accordingly. Some objects are additionally labeled with an identifier. Regions with gray hatching are not considered in this study. **b** Decomposition of the intensity (*I*) for the summer VPD extremes labeled A to I in (**a**). NH and SH refer the Northern and Southern Hemisphere. Red, dark blue, light blue, and gray bars denote contributions to *I* from positive temperature (*T*) anomalies ($I_T$), *T*-related specific humidity (*q*) anomalies ($I_{qT}$), dynamically induced *q* anomalies ($I_{qd}$), and a residual (*res*), respectively (see "Methods" for detailed description of the individual contributions).

where they were eventually hit by extreme VPD in the subsequent year of 2012, resulting in a total loss of 3.5 million head in Oklahoma and Texas[33]. Also vast regions of Europe were struck by summer VPD extremes in the last two decades (Fig. 1a), which impaired vegetation vitality (measured by vegetation indices) and reduced carbon uptake (e.g., in 2003[34], 2015[35], 2018[20,36], and 2022[6]). Also, these extremes were associated with unseen wildfires (e.g., in 2007[37], 2018[38], and 2022[2]), and substantial crop losses (e.g., in 2018[39]). The so-called "angry summer" 2018/2019 in Australia marked the hottest on record[40], during which the most intense summer VPD extreme of the southern mid-latitudes was observed (mean VPD anomaly of 1.16 kPa; Fig. 1b). Also several regions of the Global South were struck by summer VPD extremes, with particularly devastating consequences: In Argentina, the summer VPD extreme in 1988/1989 reduced the cultivated area by 35%[41], and that in 2022/2023 diminished annual GDP by an estimated 3% due to lower crop yields, livestock, and forestry activity[42]. Moreover, summer VPD extremes have been suggested to be at the root of a water conflict between Turkey and Syria in 2000[43] and allegedly led to high mortality rates and migration of wildlife in several South African national parks in summer 1982/1983[44]. In summary, the socio-ecological relevance of summer VPD extremes is obvious, however, it is currently not well understood to what extent and through which mechanisms these events are affected by climate warming. In this study we thus identify summer VPD extremes (i.e., the extremely large values) in reanalysis and climate model data as spatial objects (see "Methods") and examine changes in their intensity as the climate warms.

Climatological mean VPD has been increasing in the past four decades and is expected to rise further throughout the twenty-first century[16]. That is, global warming causes a nonlinear increase of $e_s$ of 7% per Kelvin[12], while the response of $e_a$ is in many regions less than 7%/K due to limited atmospheric moisture transport and land evapotranspiration[13,14,45]. The past increase in mean VPD and its impact on wildfire has recently been attributed directly to the emissions of major carbon producers[46]. Also for heat and drought[47], both strongly

coupled to large VPD, as well as their compound occurrence[48] significantly increased risks of extreme events to occur have been investigated in detail and attributed to global warming. However, changes of summer VPD extremes, which add to already considerable changes in mean VPD, have not been systematically assessed yet. In light of the devastating impacts of summer VPD extremes this research gap should urgently be addressed.

Here we apply an identification algorithm for seasonal VPD extremes to both reanalysis and climate model data. This algorithm, which has previously been employed to identify seasonal temperature, wind and precipitation extremes[49,50], consists of the following steps: First, an appropriate statistical model is fitted to the detrended seasonal mean VPD values at each grid point. Then, locally very rare (i.e., extreme) seasonal VPD values are identified as values exceeding the local 40-year return level. Finally, the algorithm produces spatial extreme event objects by connecting grid points that feature locally extreme VPD values in a particular season and year (see Fig. 1 for examples of spatial seasonal VPD extremes objects and "Methods" for more details on the identification algorithm). We first use data from ERA5 reanalyses[51] as well as from the Community Earth System Model version 2 large ensemble (CESM2-LENS)[52] for an "evaluation period" (1979–2023, the respective CESM2 data is hereafter referred to as CESM2[eval]) to show that the CESM2 model[53] realistically reproduces the *T* and *q* contributions to the identified summer VPD extremes. We then address the effect of climate warming on VPD extremes by contrasting CESM2-LENS summer VPD extremes identified in the 1991–2020 and 2071–2100 periods (hereafter referred to as CESM2[hist] and CESM2[eoc] datasets, respectively). Overall, we tackle the following three research goals: (1) analyze spatial patterns of the *T* and *q* contributions to the intensity of summer VPD extremes (see also Fig. 1b), (2) quantify by how much the intensity of very rare summer VPD extremes changes under global warming on top of the changes in summer mean VPD, and (3) quantify the contributions of the dominant physical processes to these changes.

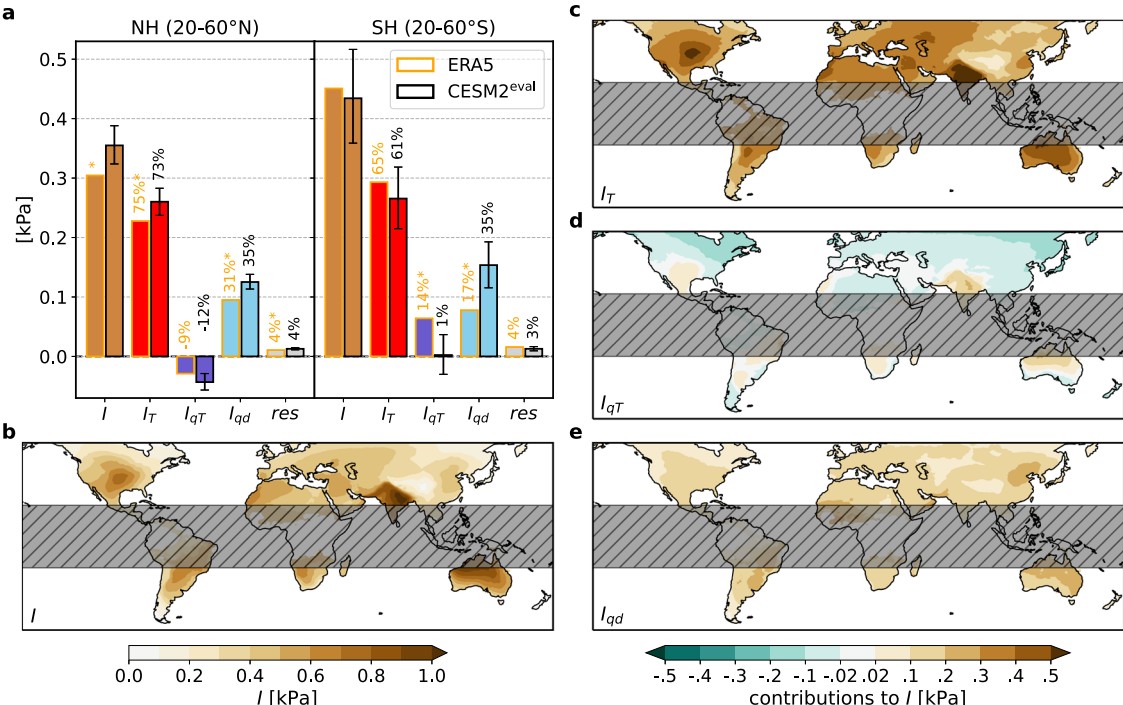

**Fig. 2 | Meteorological decomposition of intensity ($I$) in the evaluation period (1979–2023). a** The decomposition of $I$ averaged over all summer vapor pressure deficit extremes (VPD$_{S+}$), separately for the extratropical northern and southern hemisphere (NH and SH), for ERA5 (orange framed bars) and CESM2$^{eval}$ (black framed bars). CESM2$^{eval}$ values are computed as the mean over 1000 samples of equal size (number of VPD$_{S+}$ in ERA5), with whiskers showing the resulting 95% confidence interval. Values above the bars indicate the respective share of $I$, with asterisks marking statistically significant differences between ERA5 and CESM2$^{eval}$ (5% significance level). The mean intensity ($I$; **b**), its contribution from temperature anomalies $T'$ ($I_T$; **c**), from the climatological co-variability between temperature and specific humidity ($I_{qT}$; **d**), and dynamically induced humidity anomalies $q'$ ($I_{qd}$; **e**) of all VPD$_{S+}$ in CESM2$^{eval}$. Thereby the mean value at every grid cell is calculated over all VPD$_{S+}$ that comprise the respective grid cell.

## Results

### Historical summer VPD extremes

We identify 139 and 29 summer VPD extremes (VPD$_{S+}$) that affect at least $10^5$ km² land area with a center of mass in the northern (20–60°N; NH) and southern mid-latitudes (20–60°S; SH) in the ERA5 reanalysis, respectively. Figure 2a summarizes the average intensity ($I$; land average detrended VPD anomaly) over all these VPD$_{S+}$, and the shares of $I$ attributed to three potential sources of positive VPD anomalies (also see Eq. (13) in the "Methods"): warm temperature anomalies ($T' > 0$; contribution $I_T$), $T$-related deficits in atmospheric moisture ($q'_T < 0$; contribution $I_{qT}$), and negative anomalies in atmospheric moisture arising from atmospheric dynamics ($q'_d < 0$; contribution $I_{qd}$):

$$I = I_T + I_{qT} + I_{qd} + res \qquad (1)$$

The $q'_T$ are estimated from an exponential regression of $q$ with predictor $T$, hence quantifying the climatologically expected response of $q$ to a given $T$ (see "Methods"). In contrast, $q'_d$ are not related to $T$ but are the result of anomalous horizontal/vertical moisture transport and other sources of anomalous moisture. Such sources include, for example, legacies of previous rain deficits/surpluses that are mediated via soil moisture anomalies. The variability in these quantities across VPD$_{S+}$ is depicted in Supplementary Fig. 1a.

In both hemisphere's mid-latitudes, the largest share of the positive VPD anomalies during VPD$_{S+}$ is due to positive temperature anomalies. In the NH, $I_T$ contributes on average 75% to the intensity $I = 0.30$ kPa. Moreover, warm anomalies indirectly offset 9% of $I$ via the $I_{qT}$ term. This implies that, on average in the NH, a moistening occurs during warmer than usual summers. Hence, dynamically induced

moisture anomalies are another important source of 31% of $I$ in ERA5 VPD$_{S+}$. The VPD$_{S+}$ in the SH are by 0.15 kPa more intense than those in the NH due to larger contributions $I_T$ and $I_{qT}$ (Fig. 2a). That is, warm anomalies directly account for $I_T = 65\%$ of $I$ and indirectly for another $I_{qT} = 14\%$, which results from the negative $q$-$T$-correlation in many SH regions in summer (see Supplementary Fig. 2a). Note that there is very large variability in the relative contributions of these terms to $I$ across individual events, with $I_T$ ranging from 26% to 203% and $I_{qd}$ ranging from −14% to 89%.

The intensity of the 13,008 (NH) and 3625 (SH) VPD$_{S+}$ identified in the 4500-year CESM2$^{eval}$ dataset, as well as its contributions, largely agree with the above results from ERA5 (Fig. 2a). Good agreement is also found for the spatial pattern and magnitude of the 40-year return level of summer VPD anomalies in both datasets (Supplementary Fig. 3). Using a bootstrapping test, we find that in the NH, the overestimation of $I$ resulting from more positive $I_T$ and $I_{qd}$ in CESM2$^{eval}$ is significant at the 5% level, but the overestimation is less than 17% of the respective value in ERA5. In the SH, summer VPD extremes feature slightly smaller $I$, smaller $I_T$ and $I_{qT}$, and larger $I_{qd}$ in CESM2$^{eval}$ compared to ERA5, whereby only the biases in $I_{qT}$ and $I_{qd}$ are significant. These CESM2 biases are likely related to biases in the climatological $q$-$T$-correlations (Supplementary Fig. 2a, b; see "Methods"). Nevertheless, the level of agreement in the magnitude and composition of VPD$_{S+}$ is remarkable, in particular given (1) that ERA5 does not have dynamic vegetation, while the Community Land Model version 5 used in CESM2 does, and (2) given the significant challenge of adequately simulating the climatological $T$-$q$-correlation, which is used here to differentiate $I_{qT}$ from $I_{qd}$ (Supplementary Fig. 2). Overall, this analysis thus lends credibility to simulated changes in the magnitude of VPD$_{S+}$ with global warming.

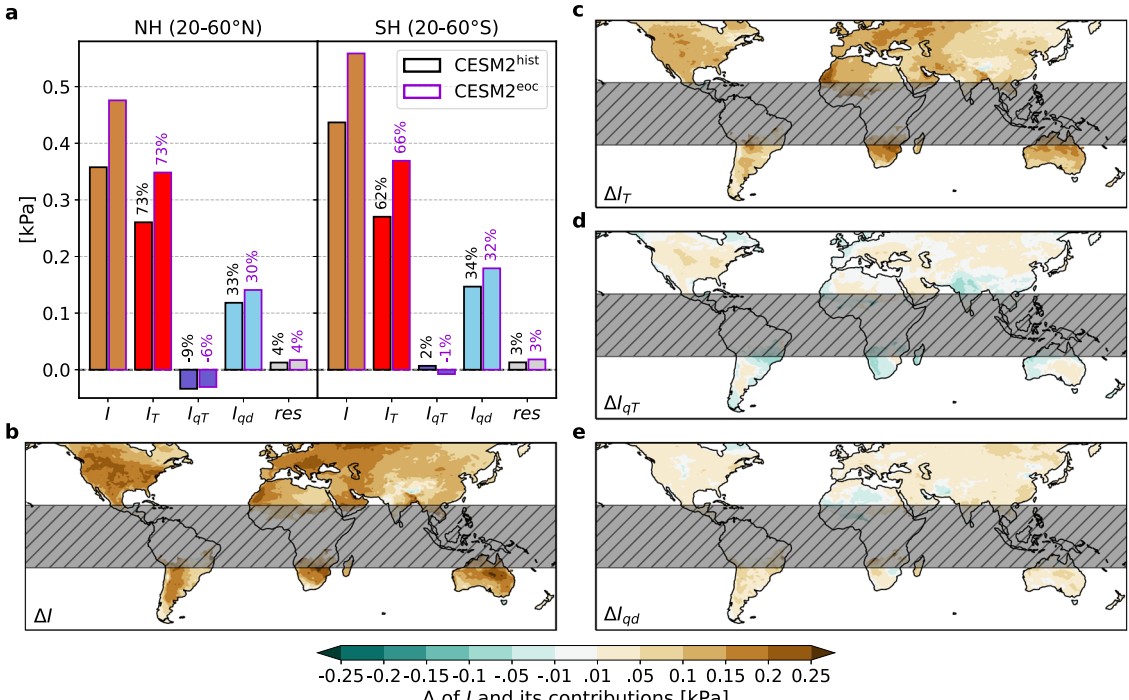

**Fig. 3 | Changes in intensity (I) between CESM2$^{hist}$ and CESM2$^{eoc}$. a** The decomposition of $I$ averaged over all summer vapor pressure deficit extremes (VPD$_{S+}$) in the northern and southern hemisphere (NH and SH) for CESM2$^{hist}$ (black framed bars) and CESM2$^{eoc}$ (purple framed bars). Values above the bars indicate the respective share of $I$. The changes in mean $I$ (**b**), in the contribution related to temperature anomalies ($I_T$) (**c**), the contribution related to the climatological co-variability between temperature and specific humidity ($I_{qT}$) (**d**), and the contribution from dynamically induced humidity anomalies ($I_{qd}$) (**e**) of VPD$_{S+}$ in CESM2$^{eoc}$ minus in CESM2$^{hist}$, whereby the mean value at every grid cell is calculated over all VPD$_{S+}$ that comprise the respective grid cell.

In the following, we assess the spatial variability of the simulated VPD$_{S+}$ in CESM$^{eval}$. The CESM$^{eval}$ offers a basis to systematically contrast characteristics of the by definition rare VPD$_{S+}$ across space (Fig. 2b–d), which is not possible based on reanalysis data alone due to too few events per grid cell. Moreover, note that the spatial variability of VPD$_{S+}$ in CESM2$^{eval}$ is almost identical to that in CESM2$^{hist}$ used in the next section, due to the large overlap of the considered periods (not shown). The most intense VPD$_{S+}$ in CESM2$^{eval}$ with $I > 0.7$ kPa occur in the continental regions of North America and in the (semi-)arid regions of Australia and southwestern Asia (Fig. 2b). This is in accordance with the hemispherically most intense observed (i.e., according to reanalysis) VPD$_{S+}$ that occurred in the U. S. in 2011 ($I = 0.78$ kPa) and in Australia in 2018/19 ($I = 1.16$ kPa; Fig. 1b). In those regions, all three meteorological anomalies contribute positively to $I$ of VPD$_{S+}$ in CESM2$^{eval}$ with the largest shares from $I_T$ (Fig. 2c–e), which was also the case for the highlighted VPD$_{S+}$ in ERA5 (Fig. 1b). At higher latitudes, e.g., in parts of Canada, South America, and Eurasia, $I_{qT}$ of VPD$_{S+}$ in CESM2$^{eval}$ is usually negative as $T$ and $q$ in summer correlate positively (Fig. 2d and Supplementary Fig. 2b). Overlapping with these regions, a rare group of VPD$_{S+}$ (accounting for 1% of all simulated VPD$_{S+}$) occurs in the Tibetan Plateau and along the lower-latitude east coasts of the continents during colder than usual summers. They hence exhibit positive $I_{qT}$, which together with dynamical dry anomalies (positive $I_{qd}$) over-compensate the negative $I_T$ (not shown). Such a combination of meteorological contributions has not been observed in ERA5, but the typically negative $I_{qT}$ and the pronounced contributions from $I_{qd}$ were also apparent in the case studies B-E and H (Fig. 1b). Generally speaking, in CESM2$^{eval}$, $I_T$ tend to be smaller over eastern portions of the continents compared to the central and western portions. Accordingly, VPD$_{S+}$ in these eastern continental regions are comparably less intense ($I \approx 0.2$ kPa) than their counterparts in the central and western continental regions. All in all, VPD$_{S+}$ in many regions acquire most of their intensity through warm anomalies but moisture-related contributions are often non-negligible.

## Intensification of VPD$_{S+}$ with global warming

Next we use the 100 member CESM2-LENS data to assess intensity changes of VPD$_{S+}$ under global warming, by comparing their $I$ and its composition between historical (hist) and end of the century (eoc) 30-year periods (1991–2020 and 2071–2100). The CESM2-LENS simulations are forced with historical forcing up to 2014 and SSP3-7.0 forcing thereafter[52]. Note that VPD$_{S+}$ are defined by rarity in their respective local climate and period. That is, a comparable number of events is identified in both the hist and eoc period. On average, VPD$_{S+}$ intensify by +33% and +28% in the NH and SH, respectively, and exhibit, on average, an intensity of $I = 0.49$ kPa across both mid-latitudes in the end-of-century period (Fig. 3a, the variability across events is depicted in Supplementary Fig. 4a). The intensification is slightly stronger in the NH showing an intensification in excess of 40% in central to northern Europe and parts of North America ($\Delta I \approx +0.2$ kPa; Fig. 3a, b). Recall that this intensification of extremes occurs on top of the changes in the climatological mean (VPD$_c$), which on average equals 0.54 kPa and 0.41 kPa in the NH and SH, respectively (shown in Supplementary Fig. 5b). The absolute 40-year return level, i.e., 40-year anomaly plus the respective VPD$_c$, increases by around 22% more strongly than VPD$_c$ alone (averaged over all regions considered here). Hence, summer VPD$_c$ will increase considerably, but seasonal VPD variability will too, with drastic consequences for VPD$_{S+}$. The intensification is dominated by changing temperature effects, i.e., increasing $I_T$ in both the NH and SH (Fig. 3a, c), although in some regions it is slightly offset or further exacerbated by changes in the $I_{qT}$ or $I_{qd}$ terms. Next, we thus investigate the physical causes of changes in $I$ (hereafter $\Delta I$) in greater detail, with a focus on changes in $I_T$ ($\Delta I_T$).

The $\Delta I_T$ (i.e., the temperature related changes in the intensity of VPD$_{S+}$) can be further decomposed into contributions from two

**Table 1 | The processes contributing to the intensification of summer vapor pressure deficit extremes (VPD$_{S+}$)**

| Definition | Process description |
|---|---|
| $\mathcal{T}_{clim} = \Delta \frac{\partial V\,PD}{\partial T}\big|_c \cdot \langle [T'] \rangle^{hist}$ | Change in the efficiency with which $T'$ generate VPD anomalies |
| $\mathcal{T}_{var} = \Delta T' \cdot \left\langle \left[ \frac{\partial V\,PD}{\partial T}\big|_c \right] \right\rangle^{hist}$ | Change in $T'$ of VPD$_{S+}$ |
| $\mathcal{Q} = \Delta I_{qT} + \Delta I_{qd}$ | $q$-related effects on $\Delta I$ |
| $\mathcal{E}$ | Residual (covariance terms, higher-order terms, and approximation used in Eq. (9)) |

Square brackets refer to averages across land grid cells contained in a given VPD$_{S+}$, and angle brackets denote averages across all VPD$_{S+}$ in one hemisphere and in the period indicated by the superscript. Note that $\Delta$-terms by definition refer to changes in mean values of VPD$_{S+}$-averaged quantities over all VPD$_{S+}$ in a particular period (see "Methods").

processes (see "Methods"): firstly, a process $\mathcal{T}_{clim}$ that quantifies the VPD anomaly change related to the non-linear dependence of VPD on $T$, i.e., changes in the efficiency with which a given $T'$ generates VPD anomalies. Secondly, changes in $T'$ during VPD$_{S+}$ themselves, termed $\mathcal{T}_{var}$. These two processes can be quantified by a linearization of the respective terms over their values in the hist period (see "Methods", Table 1 and Fig. 4). To first order, $\mathcal{T}_{clim}$ and $\mathcal{T}_{var}$ together equal $\Delta I_T$ except for minor terms that are, for ease of interpretation, attributed to a residual $\mathcal{E}$ (see "Methods"). Finally, we introduce the symbol $\mathcal{Q}$,

which denotes the sum of the changes in the two moisture related quantities $\Delta I_{qT}$ and $\Delta I_{qd}$, to arrive at a more process based decomposition of $\Delta I$ (Table 1, Eq. (15)). Hereby, the residual $\mathcal{E}$ comprises comparably small terms that are less straightforward to interpret (see Supplementary Method 1).

Figure 4 shows the process attribution of $\Delta I$ (Fig. 4a, b; see also Supplementary Fig. 6) alongside changes in the underlying physical quantities of VPD$_{S+}$ (Fig. 4c–h) and the dominant process at each grid cell (Fig. 4i). About half of $\Delta I$ in both hemispheres is due to $\mathcal{T}_{clim}$, i.e., the increasing efficiency of $T'$ to induce large VPD (Fig. 4a, b). That is, on average a $T' = 1$ K during VPD$_{S+}$ in CESM2$^{eoc}$ increases VPD by 0.23 kPa in the NH and SH, while it increases VPD only by 0.18 kPa during VPD$_{S+}$ in CESM2$^{hist}$ (Fig. 4c, f). This is due to the non-linear dependence of VPD on $T$, which strongly increases the efficiency of $T'$ to induce large VPD' as the global mean temperature increases by +3.1 K in JJA and +3.2 K in DJF. In many mid-latitude regions $\mathcal{T}_{clim}$ is the dominant process, and in arid and continental regions it is almost exclusively responsible for $\Delta I$ (Fig. 4i and Supplementary Fig. 6).

The second $T$-related process, $\mathcal{T}_{var}$, which captures changes in the $T$ variability during VPD$_{S+}$, does on average not contribute as much as $\mathcal{T}_{clim}$, but at regional scales can contribute the major share of $\Delta I$ (Fig. 4i). Averaged over all VPD$_{S+}$, $T'$ increases from +1.5 K in the NH and from +1.4 K in the SH to +1.6 K, which translates to 15% and 24% of the intensification of $I$ in the NH and SH, respectively (Fig. 4a, b). Increases in $T'$ during VPD$_{S+}$ are related to an increase in the

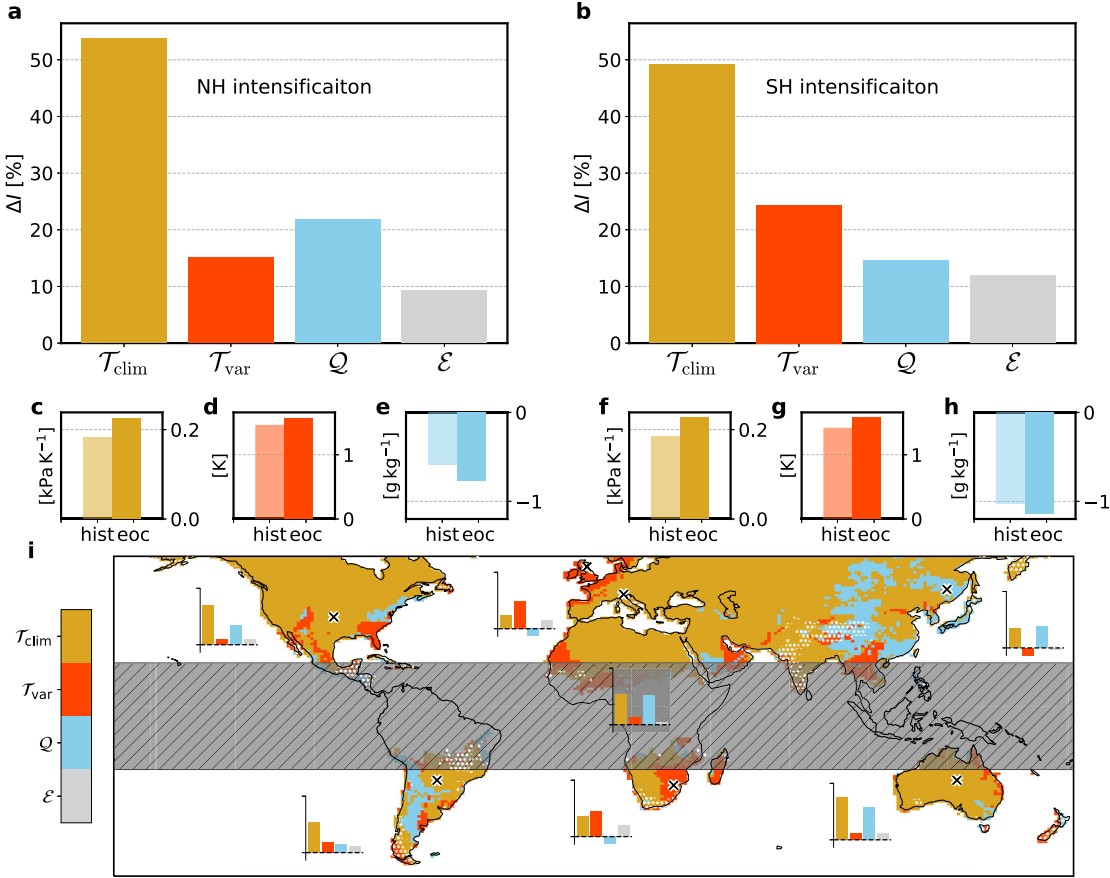

**Fig. 4 | Processes contributing to the intensification of summer vapor pressure deficit extremes (VPD$_{S+}$) between the hist and eoc period.** The mean relative shares of intensity change ($\Delta I$) contributions from the four processes in Table 1 (amber for $\mathcal{T}_{clim}$, red for $\mathcal{T}_{var}$, blue for $\mathcal{Q}$, and gray for $\mathcal{E}$) in the northern (**a**) and southern (**b**) hemisphere (NH and SH). **c–h** Values of relevant VPD$_{S+}$-averaged quantities again averaged over all VPD$_{S+}$ in CESM2$^{hist}$ (light colors) and CESM2$^{eoc}$ (darker colors) in the NH and SH, respectively. Panels depict (**c, f**) the VPD anomaly induced by $T' = 1$ K, (**d, g**) $T'$ of the respective VPD$_{S+}$, and (**e, h**) $q'$ of the respective VPD$_{S+}$. **i** The dominant process contributing to $\Delta I$ (i.e., largest contributor), whereby the attribution at every grid cell is calculated over all VPD$_{S+}$ that comprise the respective grid cell. White stippling shows grid cells with $\Delta I < 0.05$ kPa. Inset panels show the four processes as in (**a, b**) but in absolute terms at seven locations (see also Supplementary Fig. 6), which are denoted by black crosses ($y$-axis extends from 0 kPa to 0.15 kPa). See text and "Methods" for details.

interannual $T$ variability. Interestingly, in central and northern Europe as well as in parts of southeastern North America and of southern Africa, most of $\Delta I$ is attributed to $\mathcal{T}_{var}$ (Fig. 4i and Supplementary Fig. 6).

Lastly, changes in the moisture contributions from CESM2$^{hist}$ to CESM2$^{eoc}$ (summarized as $\mathcal{Q}$) contribute 22% and 15% of $\Delta I$ in the NH and SH, respectively. In many regions, the $q$-$T$-correlation becomes more negative and thus $q'$ is lower during warm summers in the end-of-century climate compared to the historical climate (see Fig. 4e, h and Supplementary Fig. 2d). This is evident in strong relative increases in $I_{qT}$ (i.e., the VPD' resulting from $T$-related $q'$) adjacent to (semi-)arid regions (Supplementary Fig. 4d). Regions that maintain a high $q$-$T$-correlation in the warmer end-of-century climate, e.g., the north-eastern continental or high-altitude regions, exert an increase in climatological mean $q$, which also increases the $q$ variability (see "Discussion"). Also there, $\mathcal{Q}$ can even be locally the most important contribution to $\Delta I$ (Fig. 4i). Note that some regions in southeastern Asia, South America, and New Zealand show a relatively weak or no intensification ($\Delta I < 0.05$ kPa; Fig. 4i). In summary, at a global scale the intensification of VPD$_{S+}$ with global warming arises mainly from $\mathcal{T}_{clim}$, i.e., the same $T'$ in a warmer climate produce more intense VPD anomalies, while a more positive $T'$ and more negative $q'$ during VPD$_{S+}$ are both adding to this intensification in a regionally differing manner.

## Discussion

This study highlights the socioeconomically highly relevant fact that very rare summer VPD extremes defined with a 40-year return period threshold (VPD$_{S+}$) intensify drastically in a warming climate. Importantly, about half of that intensification would occur already if the variability in their meteorological contributors $q'$ and $T'$ were to remain constant. This is due to exponential dependence of VPD on $T$, which implies that the same $T'$ in a warmer climate will cause larger VPD anomalies (process $\mathcal{T}_{clim}$). Larger VPD variability in a warmer climate has been documented previously in projections of VPD in the U.S.[22,54] and analyses of shorter-term VPD extremes[55]. Here, we substantially extend these studies by highlighting not only the role of mean warming separately from that of $T'$, but also by quantifying its contribution to $\Delta I$ of VPD$_{S+}$ with 54% in the NH and 49% in the SH.

Of further importance for the $\Delta I$ in both mid-latitudes are increasing $T'$ during VPD$_{S+}$ (process $\mathcal{T}_{var}$) and changes in the moisture contributions to $I$ (process $\mathcal{Q}$). The particularly strong intensification of VPD$_{S+}$ in northwestern Europe relates to the increasing interannual variability in summer $T$ (fostering $\mathcal{T}_{var}$) that has previously been related to land-atmosphere feedbacks during hot-dry summers[56,57]. Process $\mathcal{Q}$ includes mainly decreases in $q'$ during VPD$_{S+}$. On the one hand, more negative $q'$ arise from less positive $q$-$T$-correlations in a warming climate, which have been associated with the expansion of soil-moisture limited regions (in the transitional climate regime)[29], where positive $T'$ tend to occur with limited evapotranspiration and hence negative $q'$[30,31]. Such a shift from a wet to transitional climate regime is most widespread in central Eurasia and central North America[29,31], where $\mathcal{Q}$ is strongest (Supplementary Fig. 6). On the other hand, for a bounded variable like $q$ its variability increases as a result of the increase in the climatological mean $q$, as has been observed for precipitation[45,58]. In summary, both changes in interannual summer $T$ variability as well as regime-shifts in the land-atmosphere coupling also amplify the magnitude of VPD$_{S+}$.

We have shown that, at large scales, CESM2[53] is able to realistically reproduce physical characteristics of the high-impact extreme VPD summers. Even though an accurate representation of characteristics of present-day VPD$_{S+}$ does not imply that simulated changes in VPD$_{S+}$ must necessarily be accurate, this result nevertheless lends credibility to these simulated changes. The ability of GCMs to inform about future high-impact VPD$_{S+}$ has hitherto not been explored thoroughly. Evidently, the representation of VPD$_{S+}$ in GCM simulations should be

evaluated for various models, and our analyses should be repeated with other state-of-the-art GCMs and emission scenarios to corroborate our findings. However, note that even though our results were derived with only one GCM and emission scenario, we expect that the key finding of this study (VPD$_{S+}$ are intensifying drastically even relative to increasing mean VPD) is robust across models. Around 50% of this intensification is attributable to the exponential dependence of VPD on $T$, which implies that in a warmer climate, the same $T'$ yields larger VPD (process $\mathcal{T}_{clim}$)—this part of the intensification is essentially a consequence of basic physics and therefore most likely independent from the particular model that is used (albeit its magnitude might vary with the climate sensitivity of a particular model). Furthermore, at a regional scale (in particular in northwestern Europe), the pronounced increase in inter-annual summer $T$ variability is robust across different global and regional climate models[56,57]. Hence, we expect qualitatively similar $\mathcal{T}_{var}$ across GCMs, while the additional effects such as changes in summer $q$ variability or regime-shifts in the land-atmosphere coupling might be less robust across models[59–61].

Our results suggest that VPD$_{S+}$ with equal rarity will intensify drastically as the climate warms, which has severe implications for ecosystems. Put in other words, summer VPD anomalies of a given magnitude will become far more frequent in the future, and thus ecosystems not only need to adapt to larger mean values of summer VPD[16] but they also need to become resilient to larger positive anomalies of summer VPD. Given the drastic impacts of seasonal VPD extremes already in the current climate, we argue that this result should be considered by practitioners, e.g., when managing food producing systems, (inter-)national food security or forest ecosystems, or when designing agricultural reinsurance. Moreover, regarding certain impacts such as the Black Summer Wildfires in Australia in 2019/2020[1], a single VPD$_{S+}$ might have a delayed impact or one that arises only after sequential VPD$_{S+}$[1,40]. Therefore, future work should also examine consecutive or longer-than-seasonal VPD extremes so that practitioners can adequately incorporate the effects of VPD extremes in their preparedness strategies.

## Methods
### Datasets
We use hourly ERA5 reanalysis data on a horizontal 0.5° grid from December 1978 to August 2023[51]. Seasonal mean values are computed first and then bi-linearly interpolated to the CESM2 grid. Moreover, in either of the considered datasets December to February (DJF) seasons assigned to any year contain December data from the previous year. Specific humidity at 2 m, denoted as $q$, is calculated from 2-m dew point temperature and surface pressure ($p$)[62]. The 100 member CESM2-LENS[52] uses historical forcing in 1979–2014 and follows the SSP3-7.0 protocol from CMIP6[63] from 2015 onwards. It features a horizontal resolution of 1.25° longitude × -0. 9° latitude. Individual model components as well as the initialization of the 100 members is thoroughly described in Rodgers et al.[52].

To minimize effects of conditional bias[49,64] in our statistical modeling of ERA5 and CESM2 VPD data (see section "Identification of VPD$_{S+}$"), we group CESM2 data into subsets, each with the same number of years as the ERA5 data has, and then process these subsets of CESM2 data individually. That is, for identifying VPD$_{S+}$ in CESM2$^{eval}$, which consists of 100 times 45 years of data (1979–2023) each of the CESM2-LENS members is processed individually, but exactly as ERA5 data. Note that for this 45-year evaluation period, the CESM2 simulations combine historical forcing until 2014 and SSP3-7.0 forcing thereafter. For CESM2$^{hist}$ and CESM2$^{eoc}$ we combine one member (i.e., 30 years) with half of another member (i.e., 15 years) so that the resulting subset again contain 45 years of data. This procedure yields 66 subsets of 45 years (i.e., 2970 years in total) for both CESM2$^{hist}$ (in 1991–2020) and CESM2$^{eoc}$ (in 2071–2100). These subsets too are then processed exactly analogous to ERA5 data in order to identify the respective VPD$_{S+}$. Note that keeping

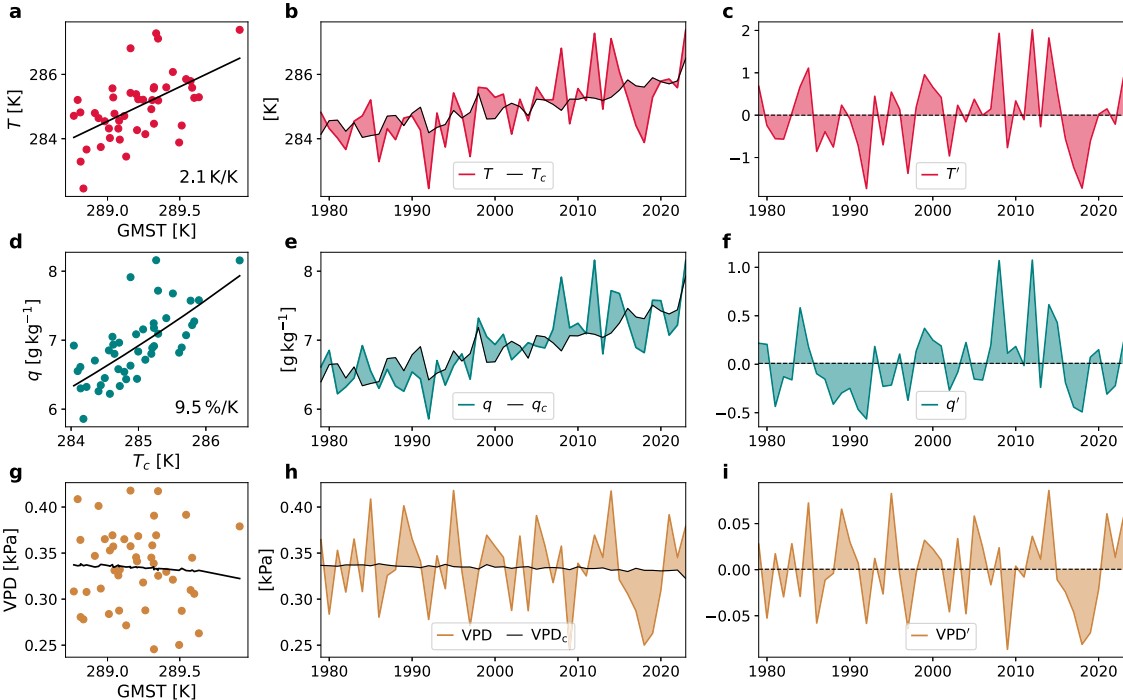

**Fig. 5 | Illustration of the forced trend estimation for ERA5 summer mean values of temperature, specific humidity, and vapor pressure deficit at the example grid cell at 54 °N, 67 °E.** Scatter plots show temperature ($T$) vs. Global mean surface $T$ (GMST) (**a**), specific humidity ($q$) vs. the forced trend estimate for $T$ ($T_c$) (**d**), and vapor pressure deficit (VPD) vs. GMST (**g**), including the respective trend curves in black. Inset numbers in (**a**, **d**) mark the slope of the trend line, which is expressed in relative terms for $q(T_c)$. **b**, **e**, **h** Evolution of the respective values (colored lines) including their trend estimate (black line) in 1979–2023. **c**, **f**, **i** Detrended anomalies, i.e., deviations from the trend line.

the number of years constant in each subset ensures comparability between the $VPD_{S+}$ in CESM2 data and ERA5, as the sample size affects the skill with which extreme events with large return periods can be identified as such through our statistical modeling[49]. We use JJA and DJF (December of the previous year) mean values in the NH and the SH, respectively, and refer to them as summer mean values. Moreover, we provide a short model evaluation based on our results and the consideration of relevant literature in the last subsection below.

**Computation and detrending of VPD**

We derive a formulation of VPD [Pa] in terms of $T$ [K], $q$ [kg kg$^{-1}$], and $p$ [Pa] starting with the definition of VPD according to Grossiord et al.[10]. For $e_s(T)$ we use Murray's formulation[65], which is valid over a wide range of $T$, and for $e_a(q, p)$ we follow the derivation by Allen et al.[66]:

$$\text{VPD} = e_s - e_a = 610.78 \cdot \exp\left(\frac{17.2694 \cdot (T - 273.16)}{T - 35.86}\right) - \frac{q \cdot p}{0.622 + 0.378 \cdot q} \quad (2)$$

Note that VPD is computed from seasonal mean values. Next, we compute detrended VPD anomalies that are physically consistent with detrended $T$ and $q$ (Fig. 5). Following Sutton et al.[67], we first compute a least square regression for local $T$ with global mean summer $T$ (GMST, computed separately in each CESM2 subset as in ERA5) as predictor (Fig. 5a):

$$T = \alpha_1 + \alpha_2 \cdot \text{GMST} + \epsilon_T = T_c + \epsilon_T \quad (3)$$

resulting in the two regression parameters ($\alpha_1$, $\alpha_2$) and $T_c$, which is a forced trend estimate for $T$, i.e., a temporally evolving climatological reference. As physics-based scaling arguments suggest an exponential relationship between $q$ and $T$[68], we estimate the forced trend in $q$ based on an exponential fit of $q$ on $T_c$ (Fig. 5d). To do so we fit a linear least square regression to $\ln(q)$ with $T_c$ as predictor. This results in the two

regression parameters $\beta_1$ and $\beta_2$, which are used to estimate the trend in $q$, i.e., $q_c$:

$$\ln(q) = \beta_1 + \beta_2 \cdot T_c + \epsilon_q \quad (4)$$

$$q_c = \exp(\beta_1 + \beta_2 \cdot T_c) \quad (5)$$

Finally, we assume no trend in $p$ with respect to GMST due to the weak dependence of VPD on $p$. Based on the time-evolving climatological values of $q$ and $T$ we compute the climatological VPD by using Eq. (2) (Fig. 5g):

$$\text{VPD}_c = \text{VPD}(T_c, q_c, p) \quad (6)$$

Deviations from the temporally evolving climatological reference values $VPD_c$, $T_c$, and $q_c$ are assumed to be entirely due to internal variability. Hence, for $X \in \{VPD, T, q\}$ the (detrended) seasonal mean anomaly ($X'$) equals (Fig. 5c, f, i):

$$X' = X - X_c \quad (7)$$

**Identification of $VPD_{S+}$**

The $VPD_{S+}$ in ERA5 and all CESM2 subsets are identified using a scheme targeted to seasonal extremes, which has previously been applied to identify seasonal extremes in $T$, surface winds, and precipitation[49,50], and consists of three steps: (1) Estimate at each grid cell the distribution of VPD' by statistical modeling and, based on that, calculate the local return period (LRP) of each VPD' value, (2) identify locally extreme VPD' based on exceedances of an LRP threshold $\tau$, and (3) form spatially coherent objects within which LRP > $\tau$ and quantify their characteristics. As VPD

has a lower bound of zero, the distributions of VPD' are typically positively skewed. To account for this skew in step (1) we fit a Yeo-Johnson transformed normal distribution ($N^{yj}$) with the three parameters $\mu$ (mean), $\sigma^2$ (variance), and $\lambda$ (transform parameter) to VPD'[69], which has previously been used to model seasonal mean $T$ and near-surface wind speed[49,50]. The three parameters are allowed to vary between grid cells and datasets, and are estimated with a maximum likelihood estimation. As for wind and $T$, also for seasonal VPD a goodness-of-fit test reveals no widespread departures of the data from the Yeo-Johnson transformed normal distribution in ERA5 and the CESM subsets (Supplementary Fig. 7). Furthermore, in steps (2) and (3), we use $\tau = 40$ years, following Boettcher et al.[50]. Hence every grid cell experiences on average about one $VPD_{S+}$ in ERA5 and in each CESM2 subset (see Supplementary Fig. 8). Further details regarding the identification of seasonal extremes are provided by the authors of the framework[49]. For our analysis we only consider $VPD_{S+}$ with a center of mass in 20–60°S (SH) and 20–60°N (NH) and a land area larger than $10^5$ km$^2$. This yields 168 $VPD_{S+}$ in ERA5, 16,633 in CESM2$^{eval}$, 11,063 in CESM2$^{hist}$, and 10,948 in CESM2$^{eoc}$.

### Intensity of $VPD_{S+}$ and its meteorological decomposition

The central aspect of every $VPD_{S+}$ studied here is its intensity ($I$), which we define as the average VPD' across all land grid cells contained in the respective $VPD_{S+}$. In the following, we decompose $I$ into contributions from $T'$, $q'_T$, and $q'_d$. To do so, we first use a multi-variable linearization of VPD from the climatological reference, i.e., we approximate any grid cell value VPD($T$, $q$, $p$) tangentially from $VPD_c = VPD(T_c, q_c, p)$[70]:

$$VPD(T,q,p) \approx VPD(T_c,q_c,p) + (T - T_c) \cdot \frac{\partial VPD}{\partial T}\bigg|_{T_c,q_c,p}$$
$$+ (q - q_c) \cdot \frac{\partial VPD}{\partial q}\bigg|_{T_c,q_c,p} \quad (8)$$

which can be formulated as:

$$VPD' = T' \cdot \frac{\partial VPD}{\partial T}\bigg|_c + q' \cdot \frac{\partial VPD}{\partial q}\bigg|_c + \epsilon_{VPD} \quad (9)$$

where $\epsilon_{VPD}$ marks a residual term containing higher-order terms, and the derivatives of VPD($T$, $q$, $p$) wrt. $T$ and $q$ are evaluated at ($T$, $q$, $p$) = ($T_c$, $q_c$, $p$) according to Eq. (2), for simplicity referred to with "c". In a second step, we account for the climatological co-variability of $T$ and $q$. That is, in some regions warmer than usual summers relate to a limited moisture supply ($q' < 0$ when $T' > 0$), and in others moistening results from the increased water holding capacity ($q' > 0$ when $T' > 0$). To estimate the climatological co-variability of $T$ and $q$, we compute an exponential regression for $q$ with predictor $T$:

$$\ln(q) = \gamma_1 + \gamma_2 \cdot T + \epsilon_{qT} \quad (10)$$

$$q_T = \exp(\gamma_1 + \gamma_2 \cdot T) \quad (11)$$

So the regression parameters $\gamma_1$ and $\gamma_2$ are used to compute the climatologically expected $q$ given $T$ ($q_T$). Again the exponentiated regression structure is chosen due to the exponential dependence of $q$ on $T$ derived from physical scaling arguments (i.e., Clausius-Clapeyron relation). Next, we rewrite the expression for the $q$-anomaly ($q'$) as

$$q' = q - q_c = (q - q_T) + (q_T - q_c) = q'_d + q'_T \quad (12)$$

that is, $q'$ is the sum of the deviation of the actual $q$ from the expected $q$ for a given temperature ($q_T$) plus the difference of $q_T$ minus the climatological $q$. The latter difference is what we refer to as $q'_T$, i.e., the $q'$ resulting from the climatological co-variability with $T$. We assume the

remaining part of $q'$, i.e., $q'_d = q - q_T$, to be the result of atmospheric dynamics and land-atmosphere interactions.

To obtain the decomposition of $I$ for a single $VPD_{S+}$, we combine Eq. (9) with Eq. (12) and average over all land grid cells contained in that $VPD_{S+}$ (denoted by square brackets):

$$I = [VPD'] = \left[T' \cdot \frac{\partial VPD}{\partial T}\bigg|_c\right] + \left[q'_T \cdot \frac{\partial VPD}{\partial q}\bigg|_c\right] + \left[q'_d \cdot \frac{\partial VPD}{\partial q}\bigg|_c\right]$$
$$+ [\epsilon_{VPD}] = I_T + I_{qT} + I_{qd} + res \quad (13)$$

where the four terms on the r.h.s. quantify the impact of each meteorological anomaly on $I$ in [Pa]. Note that in the main text, we use the term $I$ and its contributions when referring to the intensity of a single $VPD_{S+}$ and to the average over all $VPD_{S+}$ in one hemisphere. The context of the notation thus specifies whether $I$ and its contributions refer to a single (e.g., in Fig. 1b) or multiple $VPD_{S+}$ (e.g., in Fig. 2).

### Process attribution of changes in I

A common approach to attribute temporal changes in a given quantity to its components is based on a linearization of the respective relationship[71]. We apply this framework to the change in $I_T$ averaged over all $VPD_{S+}$ in one hemisphere between the historical and end-of-century period. First, we contrast the terms in Eq. (13) between the two periods:

$$\Delta I = \langle I \rangle^{eoc} - \langle I \rangle^{hist} = \Delta I_T + \Delta I_{qT} + \Delta I_{qd} + \Delta res \quad (14)$$

where angle brackets with superscript *eoc* and *hist* denote averages over all considered $VPD_{S+}$ in the given period. The $\Delta$-terms refer to the mean changes of a given contribution over all considered $VPD_{S+}$ between the two periods. Next, we re-write all components of $I_T$ evaluated in the end-of-century period (Eq. (13)) as the sum of the respective values in the historical period plus the respective deltas, e.g., $\langle [T'] \rangle^{eoc} = \langle [T'] \rangle^{hist} + \Delta T'$. Hence each $\Delta$-term quantifies the mean change of a quantity over all land grid cells of the considered $VPD_{S+}$ between the two periods. These steps are detailed in Supplementary Method 1 and yield a refined form of Eq. (14):

$$\Delta I = \underbrace{\Delta T' \cdot \left\langle \left[\frac{\partial VPD}{\partial T}\bigg|_c\right]\right\rangle^{hist}}_{\mathcal{T}_{var}} + \underbrace{\Delta \frac{\partial VPD}{\partial T}\bigg|_c \cdot \langle [T'] \rangle^{hist}}_{\mathcal{T}_{clim}} + \underbrace{\Delta I_{qT} + \Delta I_{qd}}_{\mathcal{Q}} + \mathcal{E} \quad (15)$$

which attributes the total $\Delta I$ to changes from the so-called processes $\mathcal{T}_{clim}$, $\mathcal{T}_{var}$, $\mathcal{Q}$, and $\mathcal{E}$. Note that $\mathcal{T}_{clim} + \mathcal{T}_{var} \approx \Delta I_T$ with differences only arising from covariance and higher-order terms, which are contained in the residual $\mathcal{E}$. All such details regarding the computation and definition of the four processes are available in Supplementary Method 1. In the main part of the study the processes are introduced more conceptually in Table 1 in accordance with Eq. (15).

### Significance assessment

We conduct a bootstrapping test to identify statistically significant differences in $I$ and its contributions between $VPD_{S+}$ in ERA5 and in CESM2$^{eval}$. The procedure draws 1000 random samples of $n$ $VPD_{S+}$ from CESM2$^{eval}$, whereby $n$ is the number of $VPD_{S+}$ under consideration in ERA5 ($n = 139$ in the NH, $n = 29$ in the SH). We test the null hypothesis that the $VPD_{S+}$ in ERA5 and CESM2$^{eval}$ stem from the same distribution of $VPD_{S+}$. The null distribution is constructed from the mean values over all random samples. The null hypothesis is rejected if the ERA5 mean value lies outside the 2.5th to 97.5th percentile range of the null distribution (significance level $\alpha = 5\%$). In that case, a statistically significant difference is identified for the respective characteristic of the $VPD_{S+}$ in ERA5 and CESM2$^{eval}$.

## Model evaluation

The key finding of this study, namely the fact that very rare summer VPD extremes intensify drastically in a warming climate is likely robust across models, as it is mostly due to basic physics, i.e., the non-linear dependence of VPD on $T$. Nevertheless, changes in $T$ variability and land-atmosphere coupling also affect the magnitude of $VPD_{S+}$ intensity changes, and may be model-dependent. A more thorough model evaluation regarding VPD is thus warranted. Although the hemispherically aggregated $I$ in $CESM2^{eval}$ is largely comparable to ERA5, and observed $VPD_{S+}$ agree with the spatial patterns identified in $CESM2^{eval}$, CESM2 reveals biases regarding some aspects of VPD. In the NH, $q'_T$ and $q'_d$ are influenced by regional biases in the $q$-$T$-correlation (Supplementary Fig. 2a, b) which partly compensate, hence resulting only in small discrepancies in the $I$ decomposition between ERA5 and CESM2 (Fig. 2a). Biases in the land-atmosphere coupling have been identified in several older, CMIP5 generation models, and are inherently coupled to $T$ biases[72]. In the SH, overestimated $I_{qd}$ likely relate to the lower $I_{qT}$ and $I_T$, since the $q$-$T$-correlation is too strong, i.e., warm summers show a weaker $T$-related drying ($q'_T$) and hence attribute more of $q'$ to $q'_d$ instead. Lastly, model mean state biases add to the identified discrepancies. Nevertheless, the future drying trend in many mid-latitude regions (Supplementary Fig. 2d) is consistent across $CMIP5^{61}$ and CMIP6 models[59,73], and their spatial patterns are in agreement with the identified $VPD_{S+}$ changes, especially regarding the processes resulting from stronger land-atmosphere feedback and decreasing soil moisture in summer. Finally, it should be noted that, no matter which model one uses, the exact meaning of simulated two-meter $T$, $q$ and VPD is not obvious in regions with tall vegetation. Here, however, we compare simulated $T$, $q$ and VPD with reanalysis data (rather than direct observations), which should be more comparable to one another than climate model data and direct observations.

## Data availability

ERA5 data displayed in Figs. 1 and 2 and in the Supplementary Information as well as CESM2-LENS data displayed in most of the figures are available from the ETH Research Collection via (https://doi.org/10.3929/ethz-b-000669046) [DOI will be activated upon publication, for now use the following link to access the repository: https://libdrive.ethz.ch/index.php/s/UlWEIrpQLZuslAd, Password: 3#GqESW27a2r], ref. 74. Furthermore, seasonal mean values of $T$, $q$ and VPD for both ERA5 as well as all CESM2 data used here are also available from the same repository, ref. 74. The hourly ERA5 data used to compute seasonal mean values can be downloaded from the Copernicus Climate Service (https://climate.copernicus.eu/climate-reanalysis) and have been described in detail in Hersbach et al.[51].

## Code availability

Python scripts to reproduce the main figures of this study are available via the ETH research collection, ref. 74. Furthermore, code to reproduce the set of $VPD_{S+}$ from the ERA5 and CESM2 seasonal mean $T$, $q$ and VPD, as well as code for reproducing the decomposition of their $I$ is also available from ref. 74.

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

## Acknowledgements

We acknowledge the European Centre for Medium-Range Weather Forecasts (ECMWF) and National Center for Atmospheric Research (NCAR) for providing all essential data for this study. We are grateful to Urs Beyerle from ETH Zürich for downloading the CESM2-LENS data. This research has been financially supported by the H2020 European Research Council (grant no. 787652) in the scope of the "INTEXseas" project (M.H., M.R.).

## Author contributions

All authors contributed to the planning of the study in the scope of the PhD thesis of M.H.[75]. M.H. and M.R. developed the methodology with contributing ideas from H.W. M.H. conducted the analyses, created the figures, and drafted a first version of the manuscript. All authors contributed to the final version of the manuscript.

## Funding

## Competing interests

The authors declare no competing interests.
