## [Peer Review File · Nature Communications]

Drastic increase in the magnitude of very rare summer-mean vapor pressure deficit extremesREVIEWER COMMENTS

Reviewer #1 (Remarks to the Author):

This manuscript mainly studies the changes of VPD from 1979 to 2020 and analyzes its causes. The authors came to an interesting conclusion. On the whole, the manuscript is well organized, the diagrams are well organized, and the results are clear and definite. However, I have some suggestions that I would like the author to improve.

1. The time scale used by the author seems to be the seasonal average, I would suggest using a smaller time scale.
2. The observed altitude of both temperature and specific humidity is 2m, but the author mainly studied low-latitude areas, including the Amazon, tropical rainforests of Africa and tropical rainforests of Southeast Asia, which are sensitive to extreme drought, but the height used by the author seems to be inappropriate. These forest areas should be blended height- usually 50m-200m, ERA5 2m temperature and specific humidity seems inappropriate.
3. The introduction is poorly organized, for example, Line23-24, where the sentence appears in an inappropriate place.
4. The correlation between extreme VPD and vegetation information such as NDVI changes or wildfire frequency should be analyzed to increase the reliability of the results.
5. The meaning of q_d should be explained when it first appears.
6. The author uses a large number of variables with different superscripts and subscripts, which makes the article less readable.
7. Some formulas (eq.17-20) can be placed in supporting information.

Reviewer #2 (Remarks to the Author):

Review of NCOMMS-23-62432: Drastic increase in the magnitude of very rare seasonal vapor pressure deficit extremes, by Hermann et al.

This manuscript shows the future intensification of vapor-pressure-deficit (VPD) extremes in the CESM1 model. This includes an evaluation of historical VPD extremes in the same model, based on ERA5, showing that CESM reasonably represents the intensity and contributions of different terms. The future intensification is then shown throughout the NH and SH extratropics, showing an intensification by about a third (more so in the NH), even after accounting for the mean increases. By decomposing the changes into temperature and moisture contributions, it is shown that the temperature contribution dominates on a global scale. In particular, the same temperature anomaly is more efficient at inducing large VPD at higher temperatures, although there are some regions where the increasing magnitudes of the temperature anomalies is the dominant term. In all, this is a very elegant physical decomposition of a very important metric of climate change that will

be of major societal impact due to the associated droughts, fires, and heat waves. I recommend that the manuscript can be published in Nature Communications, but have some suggestions for the authors to consider beforehand.

General comments

1) The accurate representation in CESM of large VPD extremes historically does not necessarily entail that the intensification of VPD under external forcing in the same model will be accurate. The evaluation of CESM in Fig. 2 is an important part of the study. But other studies often adopt the “emergent constraint” approach to evaluate GCMs’ future trends in some metric, rather than evaluating the GCMs’ historical climatology. I would suggest acknowledging somewhere, probably in the discussion, that there are likely inaccuracies in CESM’s representation of VPD changes under climate change that this historical analysis does not capture.

2) It is a shame that the newer CESM2 LENS is not used. I realize that you performed your own 70 simulations and this was likely before CESM2 LENS was released. Thus, I would not suggest rerunning these for CESM2. However, it is unclear whether the additional 70 simulations you performed are necessary in addition to the original 35 members to return robust results. Can you show that the original 35 macro members do not return robust results, and thus your additional 70 micro members are necessary? If the original 35 macro members do show the same results, then this would suggest the use of CESM2 instead of CESM1 in order to make the study as cutting edge as possible (admittedly the results are unlikely to change meaningfully). Although I will not request that you do so, you could also acknowledge that there are large ensembles of many other CMIP5 and CMIP6 GCMs for which this analysis could be repeated to express the model uncertainty in future intensification of VPD extremes.

Specific comments

- L73–74: This statement does not seem appropriately worded. I believe this should read, “...warm anomalies indirectly offset 9% of the I via the I_{qT} term...” or something similar.

- Fig. 2a. This figure might benefit from some representation of the spread in I and its contributions across events. At present there is uncertainty for the CESM results, but not for ERA5. Could this figure not just show a simple box and whiskers indicating the spread in each term among all events in the respective dataset?

- L78–79: Fig. 1b does not show this large variability across events (only among a very small sample of events). See previous comment that this information could be conveyed for the entire set of events in Fig. 2a.

- Could the comparisons between ERA5 and CESM in Fig. 2a be influenced by events not being identified over the same regions between the two datasets. Could we see a supplementary analysis demonstrating that the geographical distribution of events is not systematically different between the two. For example, is it possible that one dataset has more of a contribution from subtropical

regions, while the other has more of a contribution from mid-latitude regions?

- Fig. 3a. Same comment as for Fig. 2a. Could we see error bars showing how the contributions of different terms vary across events?

- Regarding Fig. 3, panels b—e may be more instructive if shown as fractional changes. The intensification of I in individual regions is difficult to interpret when shown as absolute values. Could these fractional changes be shown at least in a supplementary analysis (i.e., repeat Fig. 3b but for fractional changes)?

- L119–120: You should note that the intensification due to the temperature term is partially offset by the q_d term (Fig. 3e), particularly in the Northern Hemisphere.

- L128–129: Clarify that this new Q term is just the sum of the q_T and q_d terms from the previous decomposition, if I understand correctly?

- L131: Typo: “comparably”.

- Fig. 4 caption: Typo: The second “CESM^{hist}” should be “CESM^{eoc}”

- Fig. 4: It would be very helpful to have a legend for the light (historical) and dark (end-of-century) colors in panels c—h. Currently the reader has to read deep into the caption for this essential information.

- L145: “parts of Europe” could just read “northern Europe” for greater precision.

- L146: I’m not sure that southern Africa experiences this effect that is described for the other continents, i.e., the dominance of the T_{var} term. The results in this region appear too noisy to make this statement, likely because it is too small a region.

- L207: If I understand correctly, the 105 members includes your 70 micro members, in addition to the original 35 macro members? Please state this explicitly. This is necessary to understand the two sentences on L208–211, which initially I found impossible to follow.

- L255–256: Please clarify that this analysis for the NH and SH is separate from the grid cell analyses presented throughout the manuscript. In particular, the requirement of minimum land area does not apply to the grid cell analyses?

End of review

Reviewer #3 (Remarks to the Author):

Review of Hermann et al., manuscript number NCOMMS-23-62432

“Drastic increase in the magnitude of very rare seasonal vapor pressure deficit extremes”

This study investigates trends in the intensity and drivers of extremes in vapor pressure deficit (VPD). The authors compute and compare the intensity of summer VPD extremes which on average occur every 40 years at the beginning and end of the 21st century. Also the underlying drivers are investigated with a focus on temperature changes, humidity changes associated with the local temperature changes, and dynamically introduced humidity changes. The results show strong increases in the intensity of VPD extremes across all global mid-latitude regions, which are mainly driven by temperature increases.

Recommendation:

I think the paper requires minor revisions.

The topic of this study is interesting and timely, and the analysis provides novel insights. VPD as a measure for atmospheric water demand is an important but yet understudied variable for the global water and carbon cycles. In particular, VPD extremes are relevant for agriculture, ecosystems and consequently land carbon uptake and soil moisture droughts. Furthermore, VPD connects global warming to the water cycle and indicates to which extent water demand is growing given the expected changes in temperature (and humidity). This study is an important contribution in this context as it is the first one to map and to analyse the evolution of VPD extremes in a changing climate, and to show that these extremes seem to be reasonably well represented in an Earth System Model such as CESM1.2.

However, to further enhance this study maybe the authors can consider to expand their analyses in two directions:

General comments:

(1)

I like the comparison of the drivers of VPD extremes between ERA5 and CESM in Figure 2. However, this is not including the validation of spatial patterns and of trends, while these two aspects are part of the main results from the CESM-based analyses in Figures 2-4. I am aware that it is not the main focus of this paper to validate CESM but given the data and methods that you have already in place it should be straightforward to compare the spatial patterns from Figure 2b-e to respective ERA5 results, as well as to repeat the analyses from Figure 3 with ERA5 data from the 1980s and 2010s. This could add some additional confidence to the main findings from the CESM simulations, especially given that this is only one single model.

(2)

One aspect which is not considered so far but would add additional insight to the study is to illustrate how the detected increases in VPD extremes compare to the increases in mean VPD over the same time period. Maybe a map could be added showing the relative size of the changes in Figure 3b in comparison to the mean VPD changes.

I do not wish to remain anonymous - Rene Orth.

Specific comments:

line 26 and elsewhere: Maybe remind that readers that extremes = maxima

line 38: rephrase found → suggested

line 51: I feel it would help the general audience to add 2-3 sentences on how VPD extremes are identified (which is nicely described in detail in the Methods part)

lines 60-61: Why did you decide to limit this analysis to the mid-latitudes?

lines 81-82: Add the information that ERA5 does not include dynamic vegetation somewhere in this paragraph, as this may affect the estimated VPD extremes in this dataset.

lines 104-107: I think these two sentences are repeating the same thing.

line 107: “acquire ... due to ...”, consider rephrasing

line 150: typo in “century”

line 205: change “separate” → “additional”

line 232: There is no log-scale in Figure. Please clarify.

Figure 3: Maybe consider to add an additional version of Figure 3 showing percentage changes to the supplement.

Figure 5: Panels g-i suggests that VPD' is calculated similar to T' while line 238 states that it is calculated differently. Please clarify.

Table 1, caption: "... in one periods" —> "...in a particular period"

**Reviewer 1**

**Summary**

This manuscript mainly studies the changes of VPD from 1979 to 2020 and analyzes its causes.
The authors came to an interesting conclusion. On the whole, the manuscript is well organized,
the diagrams are well organized, and the results are clear and definite. However, I have some
suggestions that I would like the author to improve.

Thank you for this overall very positive assessment of our work.

**Comments**

1. The time scale used by the author seems to be the seasonal average, I would suggest
using a smaller time scale.

We do admit that the exact choice of time scale is always to some degree subjective,
and it would have been interesting as well to study VPD extremes on shorter time
scales. However, in our opinion the seasonal time scale is particularly interesting
when examining VPD extremes for the following reasons: (1) It is relevant from an
ecosystem perspective, as many ecosystems such as forest ecosystems react strongly
to heat and drought on seasonal time scales, but less so to heat and drought on shorter
time scales, see e.g., Fig. 3 in Vogel et al. (2021), and also Williams et al. (2013),
Senf et al. (2020), and Hermann et al. (2023). (2) As detailed on lines 26–40 (now
lines 25–44) and Fig. 1, numerous examples for seasonal VPD extremes exist during
which large impacts were observed. This motivates the systematic study of VPD
extremes on the seasonal time scale. (3) On a rather general level, we feel that
impacts and causes of extremes on the seasonal time scale are understudied, as they
are at the interface between weather and climate. With this study we aim to
contribute to research at the weather-climate interface, by focusing on one type of
seasonal extremes that is (a) particularly impactful and (b) projected to intensify in
magnitude beyond what has hitherto been acknowledged.

2. The observed altitude of both temperature and specific humidity is 2m, but the
author mainly studied low-latitude areas, including the Amazon, tropical rainforests
of Africa and tropical rainforests of Southeast Asia, which are sensitive to extreme
drought, but the height used by the author seems to be inappropriate. These forest

areas should be blended height- usually 50m-200m, ERA5 2m temperature and
specific humidity seems inappropriate.

Indeed, the meaning of 2m variables in reanalysis and climate model data is
ambiguous over regions with tall vegetation, such as tropical forests. Note, however,
that we explicitly exclude tropical regions from our analyses in this study: The
Amazon, tropical Africa, and the majority of tropical rainforests in Southeast Asia
lie outside the study region. Thus, we are somewhat bewildered by your statement
“the author mainly studied low-latitude areas”.

Nevertheless, your comment is interesting as, 2m temperature and specific humidity
values in extratropical regions with tall vegetation are subject to the same issue.
Note however, that numerous previous studies on VPD used 2m values (e.g., Yuan
et al, 2019; globally; Williams et al., 2014; over the U.S.), and our study is meant to
contribute to that body of literature. Note further that several studies on VPD also
used observational data sets such as PRISM (Williams et al. 2013, 2014; Seager et
al., 2015; U.S. only) and CRU TS (Hansen et al., 2022; globally), which are only
available at 2m above ground. Furthermore, 2m is the height for “surface” variables
that is commonly available from reanalysis and climate model data (e.g., CMIP data,
ECMWF as well as NOAA reanalyses etc.), and we feel that there are not many
sensitive alternatives to the use of 2m data in particular when comparing VPD
extremes simulated with climate models with observed events.

However, we do acknowledge that this common approach to studying VPD
extremes has evident weaknesses over tall vegetation and we now specifically state
the somewhat ambiguous meaning of any kind of simulated two-meter T , q and VPD
data in regions of tall vegetation in the model evaluation section (lines 350–353):
*“Finally, it should be noted that, no matter which model one uses, the exact meaning*
*of simulated two-meter T , q and VPD is not obvious in regions with tall vegetation.*
*Here, however, we compare simulated T , q and VPD with reanalysis data (rather*
*than direct observations), which should be more comparable than climate model*
*data and direct observations.”*

3. The introduction is poorly organized, for example, Line23-24, where the sentence
appears in an inappropriate place.

We have moved the sentence that was originally on lines 23–24 to the next
paragraph and now more clearly delineate the research gap that is being addressed
in this study (now lines 42–44).

4. The correlation between extreme VPD and vegetation information such as NDVI
changes or wildfire frequency should be analyzed to increase the reliability of the
results.

We assume the reviewer refers to the examples of high-impact seasonal VPD
extremes we discuss in the introduction (otherwise, the analyses of wildfire activity
and NDVI is out of scope of this study). To stress the impact of VPD as suggested
by the reviewer, we now more explicitly highlight the findings of the studies that
linked extreme summer VPD with vegetation information:

• Lines 27–29 now read: “The summer VPD extreme in 2011 was the most
intense in the northern mid-latitudes (mean VPD anomaly of 0.78 kPa as
122 per our definition, see Methods; Fig. 1b) *and was directly linked* to a record-
123 extensive wildfire season in the southern U.S. (Williams et al., 2014).”

• Lines 31–33 now read: “Also vast regions of Europe were struck by summer
VPD extremes in the last two decades (Fig. 1a), which impaired vegetation
vitality (*measured by vegetation indices*) and reduced carbon uptake (e.g.,
in 2003 (Ciais et al., 2005), 2015 (van Lanen et al., 2016), 2018 (Buras et
al., 2020; Senf et al., 2021), and 2022 (Van der Woude et al., 2023)).”

We believe the general socio-economic importance of VPD extremes is
substantiated to a sufficient degree by previous literature (see references on lines
25–44). Moreover, while the analysis suggested by the reviewer is certainly
interesting, we consider such a systematic assessment of extreme VPD and NDVI
or wildfire activity outside the scope of this study, also given the limited publication
space in Nature Communications. We therefore refrain from extending this study
with additional systematic analyses on the co-occurrence of VPD extremes and
NDVI or wildfire activity.

5. The meaning of qd should be explained when it first appears.

The symbol I_{qd} first appears on lines 75–76, where we now write “..., and negative
anomalies in atmospheric moisture arising from atmospheric dynamics ($q'_d < 0$;
contribution I_{qd})”. We feel that this qualifies as “explanation on first appearance”.

6. The author uses a large number of variables with different superscripts and
subscripts, which makes the article less readable.

Indeed, we see the validity of this comment. However, given that we introduce partly
novel methodologies, we feel it is important to ensure the reproducibility of our
work and thus want to document clearly what exactly we did. Unfortunately, some
terminology and symbols are unavoidable for reaching this goal and, moreover, the
use of symbols considerably shortens and simplifies sentences throughout the
manuscript. However, to improve the clarity of our notation we went through the
manuscript and added plain word explanations to symbols to remind the reader of
their meaning, wherever we felt that this was appropriate.

7. Some formulas (eq.17-20) can be placed in supporting information.

We cannot quite follow your comment here. The main paper has only 15 equations,
and Eqs. 16–19 are already in the Supplementary Information. There is no Eq. 20,
neither in the original submission nor in the revised version of this manuscript.

**Reviewer 2**

**Summary**

This manuscript shows the future intensification of vapor-pressure-deficit (VPD) extremes in
the CESM1 model. This includes an evaluation of historical VPD extremes in the same model,
based on ERA5, showing that CESM reasonably represents the intensity and contributions of
different terms. The future intensification is then shown throughout the NH and SH extratropics,
showing an intensification by about a third (more so in the NH), even after accounting for the
mean increases. By decomposing the changes into temperature and moisture contributions, it is
shown that the temperature contribution dominates on a global scale. In particular, the same
temperature anomaly is more efficient at inducing large VPD at higher temperatures, although
there are some regions where the increasing magnitudes of the temperature anomalies is the
dominant term. In all, this is a very elegant physical decomposition of a very important metric
of climate change that will be of major societal impact due to the associated droughts, fires, and
heat waves. I recommend that the manuscript can be published in Nature Communications, but
have some suggestions for the authors to consider beforehand.

Thank you for your supportive and thorough review of our manuscript. Your comments very
much helped to improve the presentation of our work and led to several additional and insightful
analyses.

**General remarks**

1. The accurate representation in CESM of large VPD extremes historically does not
necessary entail that the intensification of VPD under external forcing in the same
model will be accurate. The evaluation of CESM in Fig. 2 is an important part of
the study. But other studies often adopt the “emergent constraint” approach to
evaluate GCMs’ future trends in some metric, rather than evaluating the GCMs’
historical climatology. I would suggest acknowledging somewhere, probably in the
discussion, that there are likely inaccuracies in CESM’s representation of VPD
changes under climate change that this historical analysis does not capture.

We agree with this reviewer comment and have therefore re-written the paragraph
in the discussion section where we allude to potential caveats of our study. Also, the
paragraph has been adjusted to the usage of CESM2-LENS data as opposed to the
CESM1 simulations presented in the original submission of this manuscript. The
paragraph now reads (lines 198–212): “We have shown that, at large scales,
CESM2⁵³ is able to realistically reproduce physical characteristics of the high-

*impact and very rare VPD_{S+}. Even though an accurate representation of*
*characteristics of present-day VPD_{S+} does not imply that simulated changes in*
*VPD_{S+} must necessarily be accurate, this result nevertheless lends credibility to*
*these simulated changes. The ability of GCMs to inform about future high-impact*
*VPD_{S+} has hitherto not been explored thoroughly. Evidently, the representation of*
*VPD_{S+} in GCM simulations should be evaluated for various models, and our*
*analyses should be repeated with other state-of-the-art GCMs and emission*
*scenarios to corroborate our findings. However, note that even though our results*
*were derived with only one GCM and emission scenario, we expect that the key*
*finding of this study (VPD_{S+} are intensifying drastically even relative to increasing*
*mean VPD) is robust across models. Around 50% of this intensification is*
*attributable to the exponential dependence of VPD on T, which implies that in a*
*warmer climate, the same T' yields larger VPD (process τ_{clim}) – this part of the*
*intensification is essentially a consequence of basic physics and therefore most*
*likely independent from the particular model that is used (albeit its magnitude might*
*vary with the climate sensitivity of a particular model). Furthermore, at a regional*
*scale (in particular in northwestern Europe), the pronounced increase in inter-*
*annual summer T variability is robust across different global and regional climate*
*models^{55, 56}. Hence, we expect qualitatively similar T_{var} across GCMs, while the*
*additional effects such as changes in summer q variability or regime-shifts in the*
*land-atmosphere coupling might be less robust across models^{58–60}.”*

- 2. It is a shame that the newer CESM2 LENS is not used. I realize that you performed
your own 70 simulations and this was likely before CESM2 LENS was released.
Thus, I would not suggest rerunning these for CESM2. However, it is unclear
whether the additional 70 simulations you performed are necessary in addition to
the original 35 members to return robust results. Can you show that the original 35
macro members do not return robust results, and thus your additional 70 micro
members are necessary? If the original 35 macro members do show the same results,
then this would suggest the use of CESM2 instead of CESM1 in order to make the
study as cutting edge as possible (admittedly the results are unlikely to change
meaningfully). Although I will not request that you do so, you could also
acknowledge that there are large ensembles of many other CMIP5 and CMIP6

GCMs for which this analysis could be repeated to express the model uncertainty in
future intensification of VPD extremes.

We agree with your comment and decided to repeat all our analyses with the newer
CESM2-LENS data, for which 100 members are publicly available by now.
Evidently, this comment led to major changes in our study setup, and some
quantitative changes in our results. However, qualitatively, all main conclusions of
this work proved to be robust to this switch to a newer model and different radiative
forcing. Specifically, in the resubmitted version of the manuscript we used the
CESM2-LENS data in the following way:

- For the model evaluation we used the full 100 members for the 1979–2023 and
hereafter (and in the revised manuscript) refer to this dataset as CESM2^{eval}. Recall
that in the original submission we used a 105 member CESM1 ensemble for the
period 1990–2000. Hereby, the larger temporal overlap between CESM2-LENS and
ERA5 now allows for a more robust evaluation. Compared to CESM1, the
overestimation of the intensity of VPD_{S+} is now somewhat larger in CESM2 (in
particular in the NH). Qualitatively, however, the mean composition of the I of
VPD_{S+} and its spatial patterns remain unaltered, perhaps except for I_{qT} in the NH
midlatitudes, which is more negative for CESM2 compared to CESM1 (new Fig. 2).
The more negative I_{qT} there results from more positive correlations between T and
q in these regions in CESM2 compared to CESM1.

- To assess the climate change effect on the intensity of simulated VPD_{S+}, we now
select two 30-year periods for comparison – a “historical” period 1991–2020 and an
end-of-the-century period 2071–2100 – and refer to the respective CESM2 datasets
as CESM2^{hist} and CESM2^{eoC} (hereby the 1991–2020 period corresponds to the
current “climatological standard normal period” of the WMO). Recall that in the
original submission we compared simulations for the 1990–2000 and 2091–2100
periods. As a consequence, changes in the intensity of VPD_{S+} are now somewhat
smaller (+33% and +28% in the NH and SH, respectively compared to +54% and
+31% in the original submission), for two reasons: (a) The CESM2-LENS
simulations use SSP3–7.0 forcing, while the simulations used in our original
submission used RCP8.5 forcing, i.e., CESM2-LENS data see a 1.5 W m^{-2} weaker
increase in the radiative forcing by the end of the century compared to the

simulations used in our original submission. (b) We compare time periods that are
closer together in time. Nevertheless, these increases are still drastic and, moreover,
their attribution to different processes remains qualitatively unchanged. The VPD_{S+}
intensification attributable to the non-linear derivative of VPD with respect to T
(process J_{clim}) still accounts for $\sim 50\%$ of these increases in either hemisphere, and
the spatial pattern of the dominant process in the intensification of VPD_{S+} remains
very similar too.

Overall, we thus believe that the exercise of switching to CESM2-LENS for this
analysis was well worth the effort, as it increases our confidence in our results, even
though it meant reproducing all figures of this manuscript except Fig. 1. We are thus
grateful to the reviewer for this comment!

**Specific comments**

3. L73–74: This statement does not seem appropriately worded. I believe this should
read, “...warm anomalies indirectly offset 9% of the I via the I_{qT} term...” or
something similar.

Yes, indeed that was unclear. Changed as you suggested (now on lines 85–86).

4. Fig. 2a. This figure might benefit from some representation of the spread in I and its
contributions across events. At present there is uncertainty for the CESM results,
but not for ERA5. Could this figure not just show a simple box and whiskers
indicating the spread in each term among all events in the respective dataset?

We have added a plot displaying the variability in the VPD_{S+} 's I and its contribution
as new Fig. S1a.

5. L78–79: Fig. 1b does not show this large variability across events (only among a
very small sample of events). See previous comment that this information could be
conveyed for the entire set of events in Fig. 2a.

Indeed, Fig. 1b only considers a small sample of events, but at least for this sample
it does show considerable variability across events, e.g., for the event A (NH) the I
consists of 70% I_T and 7% I_{qd} , while for the event E (NH), the I_T contribution is only
35% but the I_{qd} contribution is 71%. As noted in our reply to your specific comment

4, we have now included Fig. S1a in the supplement to visualize this variability
across events.

6. Could the comparisons between ERA5 and CESM in Fig. 2a be influenced by events
not being identified over the same regions between the two datasets. Could we see
a supplementary analysis demonstrating that the geographical distribution of events
is not systematically different between the two. For example, is it possible that one
dataset has more of a contribution from subtropical regions, while the other has more
of a contribution from mid-latitude regions?

This is indeed a somewhat tricky question that requires some elaboration, but the
short answer is no. Per design of our analysis, it is highly unlikely that this is the
dominant cause of discrepancies between ERA5 and CESM2 VPD extremes. Note
that the events are defined based on their rarity (i.e., 40-yrs local return period). This
means that, per definition, there cannot be a systematic difference in their
geographical distribution as in both data sets, roughly one event per 40 years is
expected at each grid point.

However, it is nevertheless entirely expected from basic statistical considerations
that some regions in ERA5 feature more than one 40-year VPD extreme in 45 years,
while in other regions no such event occurs in the same period at all. The situation
is essentially analogous to rolling a dice six times – there too it might happen that
within six rolls you roll a six several times or not at all. These discrepancies in the
number of events per grid point are, however, solely due to random sampling
uncertainty and arise because we only sample a relatively short period in ERA5.
Figure S8 shows that indeed some ERA5 grid points are covered by two 40-year
VPD extremes in the 1979–2023 period, while at other grid points no 40-year event
occurred in that period. Also, note how spatially homogenous the fields depicted in
Fig. S8b–d are compared to those in Fig. S8a.

If one compared the characteristics of the (small sample of) ERA5 events to those
of the (much larger sample of) CESM2 events, then indeed that comparison might
be affected by the much larger sampling uncertainty in ERA5 compared to CESM2.
This issue is exactly what motivated us to perform the resampling test that resulted
in the whiskers on top of the CESM2 bars in Fig. 2a (detailed on lines 328–335). To

account for differences in sampling uncertainty in CESM2 and ERA5 (i.e., to assess
for which characteristics sampling uncertainty can be ruled out as cause of
differences between CESM2 and ERA5 VPD extremes), we thus selected 1000
330 times the same number of CESM2 VPD_{S+} events as we have in ERA5, and then
compute the respective quantities shown in Fig. 2a for these subsamples of CESM2
events. These CESM2 subsamples then all suffer from the same sampling
uncertainty as the ERA5 sample and thus the distributions of their respective
characteristics, I , I_T , etc. (which are depicted with the whiskers on top of the CESM2
bars in Fig. 2a) then allow assessing whether differences between the characteristics
of CESM2 and ERA5 events could result solely from this sampling uncertainty.
Essentially, we test the null-hypothesis that the CESM2 and ERA5 VPD extremes
all come from the same distribution of VPD extremes and we reject that null-
hypothesis (i.e., we identify a significant differences in CESM2 and ERA5 VPD
extremes) when the value from the ERA5 sample lies outside the 2.5th to 97.5th
percentile range (shown as whiskers on top of the CESM2 bars in Fig. 2a) of the
respective values from the 1000 CESM2 subsamples.

7. Fig. 3a. Same comment as for Fig. 2a. Could we see error bars showing how the
contributions of different terms vary across events?

We have added Fig. S4a which now depicts the variability across events in
CESM2^{hist} and CESM2^{roc}.

8. Regarding Fig. 3, panels b—e may be more instructive if shown as fractional
changes. The intensification of I in individual regions is difficult to interpret when
shown as absolute values. Could these fractional changes be shown at least in a
supplementary analysis (i.e., repeat Fig. 3b but for fractional changes)?

Fractional changes are now shown in panels c—e in Fig. S4.

9. L119–120: You should note that the intensification due to the temperature term is
partially offset by the q_d term (Fig. 3e), particularly in the Northern Hemisphere.

We have added to this sentence (now lines 137–138): “..., although in some regions
it is slightly offset or further exacerbated by changes in the I_{qd} term.”

10. L128–129: Clarify that this new Q term is just the sum of the qT and qd terms from
the previous decomposition, if I understand correctly?

Yes, as detailed in Table 1, Q is the sum of the changes in I_{qT} and I_{qd} .

11. L131: Typo: “comparably”.

Yes, indeed, thanks, changed!

12. Fig. 4 caption: Typo: The second “CESM[^]hist” should be “CESM[^]eoc”

Yes, indeed, thanks, changed!

13. Fig. 4: It would be very helpful to have a legend for the light (historical) and dark
(end-of-century) colors in panels c–h. Currently the reader has to read deep into
the caption for this essential information.

Thanks for this comment. We added x-axis labels in these panels in Fig. 4 to
alleviate this issue.

14. L145: “parts of Europe” could just read “northern Europe” for greater precision.

We rephrased it to “northern Europe”.

15. L146: I’m not sure that southern Africa experiences this effect that is described for
the other continents, i.e., the dominance of the T_{var} term. The results in this region
appear too noisy to make this statement, likely because it is too small a region.

Thank you for pointing this out. We agree and we have thus removed “southern
Africa” in this sentence.

16. L207: If I understand correctly, the 105 members includes your 70 micro members,
in addition to the original 35 macro members? Please state this explicitly. This is
necessary to understand the two sentences on L208–211, which initially I found
impossible to follow.

We believe this comment is no longer relevant after switching to the CESM2-LENS
data.

17. L255–256: Please clarify that this analysis for the NH and SH is separate from the
grid cell analyses presented throughout the manuscript. In particular, the
requirement of minimum land area does not apply to the grid cell analyses?

We think there is a misunderstanding here. Throughout the manuscript we present
analyses of the spatial extreme VPD objects, identified as detailed in the section
“Identification of VPD_{s+}”. Maps displayed in Figs. 2–4 depict at each grid point
characteristics computed as the mean over all objects that cover the respective grid
point. To make this aspect of our work clearer, we now underlined the fact that we
consider spatial objects (VPD_{s+}) very explicitly already in the introduction, on lines
55–61: *“This algorithm, which has previously been employed to identify seasonal
temperature, wind and precipitation extremes^{49,50} consists of the following steps:
First, an appropriate statistical model is fitted to the detrended seasonal mean VPD
values at each grid point. Then, locally very rare (i.e., extreme) seasonal VPD
values are identified as values exceeding the local 40-year return level. Finally, the
algorithm produces spatial extreme event objects by connecting grid points that
feature locally extreme VPD values in a particular season and year (see Fig. 1 for
examples of spatial seasonal VPD extremes objects and Methods for more details
on the identification algorithm).”*

**Reviewer 3 (René Orth)**

**Summary**

This study investigates trends in the intensity and drivers of extremes in vapor pressure deficit
(VPD). The authors compute and compare the intensity of summer VPD extremes which on
average occur every 40 years at the beginning and end of the 21st century. Also the underlying
drivers are investigated with a focus on temperature changes, humidity changes associated with
the local temperature changes, and dynamically introduced humidity changes. The results show
strong increases in the intensity of VPD extremes across all global mid-latitude regions, which
are mainly driven by temperature increases.

**Recommendation:**

I think the paper requires minor revisions.

The topic of this study is interesting and timely, and the analysis provides novel insights.
VPD as a measure for atmospheric water demand is an important but yet understudied variable

for the global water and carbon cycles. In particular, VPD extremes are relevant for agriculture,
ecosystems and consequently land carbon uptake and soil moisture droughts. Furthermore,
VPD connects global warming to the water cycle and indicates to which extent water demand
is growing given the expected changes in temperature (and humidity). This study is an
important contribution in this context as it is the first one to map and to analyse the evolution
of VPD extremes in a changing climate, and to show that these extremes seem to be reasonably
well represented in an Earth System Model such as CESM1.2.

However, to further enhance this study maybe the authors can consider to expand their analyses
in two directions:

We would like to thank the reviewer for his apparent interest in our work and for his
constructive and supporting comments, which helped greatly to improve the clarity of our work.

**General Comments**

1. I like the comparison of the drivers of VPD extremes between ERA5 and CESM in
Figure 2. However, this is not including the validation of spatial patterns and of trends,
while these two aspects are part of the main results from the CESM-based analyses in
Figures 2-4. I am aware that it is not the main focus of this paper to validate CESM but
given the data and methods that you have already in place it should be straightforward
to compare the spatial patterns from Figure 2b-e to respective ERA5 results, as well as
to repeat the analyses from Figure 3 with ERA5 data from the 1980s and 2010s. This
could add some additional confidence to the main findings from the CESM simulations,
especially given that this is only one single model.

We agree that a more extensive evaluation of CESM2 regarding VPD (extremes) further
strengthens this study. Unfortunately, however, the maps shown in Fig. 2b–e cannot be
reproduced in a meaningful way for ERA5 data, due to the small number of VPD_{S+}
events in ERA5 (0-2 events for most grid cells, see Fig. S8). Recall that these maps
show quantities computed as the mean over the respective values of all VPD_{S+} in
CESM2^{eval} that cover the respective grid point. This is only meaningful when the
respective characteristics can be averaged over a larger number of events, as is the case
for CESM2^{eval}, but not for ERA5, where typically only 0-2 VPD_{S+} occur at each grid
point. However, to evaluate CESM2 regarding VPD extremes in a more detailed and
spatially explicit manner we have added Fig. S3, which depicts the 40-year return level
of summer VPD anomalies in either of the data sets. This figure shows that the spatial

pattern in the threshold to define a VPD extreme is remarkably well captured by
CESM2. Figure S3 is thus somewhat analogous to the reviewer's wish/suggestion to
have the maps in Fig. 2 also available for ERA5, although it only considers the spatial
pattern in the magnitude (i.e., analogous to Fig. 2b) of VPD extremes..

2. One aspect which is not considered so far but would add additional insight to the study
is to illustrate how the detected increases in VPD extremes compare to the increases in
mean VPD over the same time period. Maybe a map could be added showing the relative
size of the changes in Figure 3b in comparison to the mean VPD changes.

Excellent suggestion, thanks! We have added Fig. S5, which compares the average
change in the absolute 40-year return level (panel a) to the average change in the
climatological mean VPD (VPD_c) between the hist and eoc period. Panel (c) then
depicts by how much the former exceeds the latter (in relative terms). Note that the
absolute 40-year return level denotes the threshold for summer mean VPD above which
a given seasonal VPD value is considered locally extreme, i.e., which occurs no more
than once every 40 years. The figure clearly illustrates that this threshold increases more
strongly than VPD_c , by about 22% when averaging across all regions considered in this
study.

**Specific Comments**

3. line 26 and elsewhere: Maybe remind that readers that extremes = maxima
True, we now state this clearly on line 43.

4. line 38: rephrase found → suggested

Ok, changed as suggested.

5. line 51: I feel it would help the general audience to add 2-3 sentences on how VPD
extremes are identified (which is nicely described in detail in the Methods part)

Yes, indeed in the originally submitted version it was not entirely clear from only the
main text how the identification of spatial VPD extreme objects works. We have now
added a few sentences on lines 55–61.

6. lines 60-61: Why did you decide to limit this analysis to the mid-latitudes?

Several reasons led us to that choice: (a) Our study is motivated by recent examples of
high-impact VPD extremes in extratropical regions, i.e., the ones discussed in the
introduction. As we state clearly in the introduction, the goal of this study is to
investigate how extreme events of this particular kind change under warming. (b) In
tropical regions the classical interpretation of “seasons” of roughly 3 months duration
is questionable. (c) We are not aware of high-impact VPD extremes in polar regions.
While we did identify VPD_{S+} also at high latitudes, we felt that due to their far smaller
socioeconomic impacts they should not be in the focus of our analysis, and thus chose
to not include them in the results presented here. (d) Due to the rareness of the events
we analyse, we are forced to pool events across space in order to draw statistically robust
conclusions (e.g., those shown in Figs. 2a, 3a and 4). Hereby, we gain in statistical
robustness the more events we have, i.e., the larger the region is over which we pool.
However, the larger that region the less homogenous the pooled events will be, simply
due to the diversity of climate zones and the causes of T and q anomalies across the
globe. Thus, by focusing on mid-latitude events we aimed at pooling events across
regions that exhibit at least some degree of comparability regarding the ways how the
climate system functions (e.g., how it produces T and q anomalies). In fact, we did
identify VPD extremes globally for both the JJA and DJF seasons, but then felt that
including events from the tropics to the poles would render the analyses presented in
Figs. 2a and 3a and 4 less insightful, as they would just lump together events from
regions that are climatically too diverse.

7. lines 81-82: Add the information that ERA5 does not include dynamic vegetation
somewhere in this paragraph, as this may affect the estimated VPD extremes in this
dataset.

Ok, done (now line 101).

8. lines 104-107: I think these two sentences are repeating the same thing.

Indeed, the word “however” was misleading. We have reformulated the respective
sentences to be more specific (and to account for the slight changes in the results due to
the switch from CESM1.2 to CESM2-LE). They now read: “Generally speaking, in
CESM2^{eval}, I_T tend to be smaller over eastern portions of the continents compared to the
central and western portions. Accordingly, VPD_{S+} in these eastern continental regions
are comparably less intense ($I \approx 0.2$ kPa) than their counterparts in the central and

western continental regions.” (lines 121–122). The second sentence is meant to
highlight the importance of (high) I_T for (high) I , and to explain why some regions show
comparably small values of I .

9. line 107: “acquire ... due to ...”, consider rephrasing

We rephrased to “acquire ... through ...”.

10. line 150: typo in “century”

Yes, indeed, thanks for spotting!

11. line 205: change “separate” → “additional”

Ok, changed as requested.

12. line 232: There is no log-scale in Figure. Please clarify.

That is true. But note the slight bends in the black lines in Fig. 5d,g, which result from
the exponential fit of q on T_c . Admittedly, the effect of regressing T_c on $\ln(q)$ instead of
q is minimal in this case (i.e., as long as the range of T_c values is comparatively small),
but it is justified physically, as through Clausius-Clapeyron one expects an exponential
relationship between q_c and T_c .

13. Figure 3: Maybe consider to add an additional version of Figure 3 showing percentage
changes to the supplement.

Excellent suggestion. We have added Fig. S4 which shows the relative changes (relative
to the magnitude of the respective value in CESM^{hist}).

14. Figure 5: Panels g-i suggests that VPD’ is calculated similar to T’ while line 238 states
that it is calculated differently. Please clarify.

Thanks for making us aware that the differing computations of the T , q and VPD
climatologies (T_c , q_c and VPD_c and the respective anomalies relative to these quantities)
are not sufficiently clear. In plain words what we do is the following: (1) The
temperature climatology T_c is computed at each grid point by regressing T on GMST,
whereby the prediction of the regression model ($\alpha_1 + \alpha_2 GMST = T_c$) is then considered
the temperature climatology (Eq. 3). (2) Since basic physical consideration (i.e., the
Clausius-Clapeyron relation) suggest exponentially dependent T_c and q_c , we compute q_c

by regressing the natural logarithm of the q values on the previously computed T_c values.
Again, q_c is then the prediction of that regression ($q_c = \exp(\beta_1 + \beta_2 T_c)$). This
regression is depicted exemplarily in Fig. 5d, which has T_c on its x-axis, contrary to Fig.
5a, which has GMST on the x-axis. Note that T_c and q_c computed in this way are not
independent but rather linked through an exponential function (quantifying the
percentage change in q_c per K change in T_c), which we feel is justified given the physical
expectation of exponentially dependent q_c and T_c . (3) The so computed q_c and T_c values
then allow us to compute a climatological VPD value, VPD_c , which is entirely consistent
with q_c and T_c . For this study, we consider our approach superior to computing VPD_c
simply as some (exponential or linear) regression of GMST on VPD, as in that case the
resulting VPD_c would **not** be physically consistent with the q_c and T_c . This consistency,
however, is important here as we aim to disentangle the effect of T' and q' on VPD'
which only add up if the q_c , T_c and VPD_c are physically consistent with one another.

In summary: The computation of VPD' differs from that of T' as T' is simply the
residual of a regression of GMST on T , while VPD' is the difference between the actual
observed VPD and the VPD_c computed from T_c and q_c (i.e., not the residual of a simple
linear regression of GMST on VPD).

To accommodate this comment, we have changed the example grid cell displayed in
Fig. 5 to one that clearly shows that VPD_c does not result from any kind of regression/fit.

15. Table 1, caption: "... in one periods" → "...in a particular period"

Yes, thanks, corrected!

**References**

In this document, we mostly refer to references cited in the manuscript. Hence, we refer the
reader to the reference list of the manuscript, in addition to the following references, which are
not cited in the main paper:

Hansen, W. D., Schwartz, N. B., Williams, A. P., Albrich, K., Kueppers, L. M., Rammig, A.,
Reyer, C. P. O., Staver, A. C., & Seidl, R. (2022). Global forests are influenced by the
legacies of past inter-annual temperature variability. *Environmental Research: Ecology*,
1(1), 011001. <https://doi.org/10.1088/2752-664X/ac6e4a>

Senf, C., Buras, A., Zang, C. S., Rammig, A., & Seidl, R. (2020). Excess forest mortality is
consistently linked to drought across Europe. *Nature Communications*, 11(1), 6200.
<https://doi.org/10.1038/s41467-020-19924-1>

REVIEWERS' COMMENTS

Reviewer #2 (Remarks to the Author):

Thanks to the authors for a very thorough revision. It is reassuring that similar results hold for CESM2-LENS as for CESM1-LENS. The revised manuscript reads excellently and will be of high impact.

One final very minor issue I would raise is that the period 1979-2023 is used for model evaluation, which in CESM2 combines historical (1979-2014) and SSP3-7.0 (2015-2023) forcings. Hence, comparing CESM2 and ERA5 over this period technically is not comparing like-with-like. I am not concerned that this impacts the results, and the upside of evaluating the GCM up to 2023 of course is that it enables a greater period of ERA5 to be used for GCM evaluation. So it could be worth acknowledging that the evaluation period combines the historical and SSP3-7.0 simulations, but giving the justification for doing so (this could just be in the methods). Otherwise, I recommend publication in the manuscript in its current form.

Reviewer #3 (Remarks to the Author):

Second Review of Herman et al., NCOMMS-23-62432A

“Drastic increase in the magnitude of very rare seasonal vapor pressure deficit extremes“

The authors have (care)fully addressed my comments, and with the respective additions and clarifications they included in the revised manuscript, I now fully support the publication of this paper in its present form.

I do not wish to remain anonymous - Rene Orth.

**General comments to the three reviewers**

We would like to thank the reviewers again for their very positive response to our revised
manuscript. Below we have included the very last comment by Reviewer #2 (in black)
alongside our according reply (in blue). The other reviewers did not suggest any further
modifications of the manuscript.

**Reviewer 2**

**Last remarks**

Thanks to the authors for a very thorough revision. It is reassuring that similar results hold for
CESM2-LENS as for CESM1-LENS. The revised manuscript reads excellently and will be of
high impact.

Thank you once again for this very positive feedback.

**Last comment**

One final very minor issue I would raise is that the period 1979-2023 is used for model
evaluation, which in CESM2 combines historical (1979-2014) and SSP3-7.0 (2015-2023)
forcings. Hence, comparing CESM2 and ERA5 over this period technically is not comparing
like-with-like. I am not concerned that this impacts the results, and the upside of evaluating
the GCM up to 2023 of course is that it enables a greater period of ERA5 to be used for GCM
evaluation. So it could be worth acknowledging that the evaluation period combines the
historical and SSP3-7.0 simulations, but giving the justification for doing so (this could just
be in the methods). Otherwise, I recommend publication in the manuscript in its current form.

This is indeed worth noting. We have included two sentences as suggested by the reviewer in
the method section. We modified the description of the CESM2 dataset, which now reads “The
100 member CESM2–LENS⁵² uses historical forcing in 1979–2014 and follows the SSP3–7.0
protocol from CMIP6⁶³ from 2015 onwards. It features a horizontal resolution...”. Additionally
we added a sentence in the next paragraph (describing the CESM2 model evaluation) that reads
“Note that for this 45-year evaluation period, the CESM2 simulations combine historical
forcing until 2014 and SSP3-7.0 forcing thereafter.”.